# Assimilation of GNSS tomography products into WRF using radio occultation data assimilation operator

Natalia Hanna[1], Estera Trzcina[2], Gregor Möller[1*], Witold Rohm[2], Robert Weber[1]

[1]Department of Geodesy and Geoinformation, TU Wien, Vienna, 1040, Austria

[2]Institute of Geodesy and Geoinformatics, Wroclaw University of Environmental and Life Sciences, Wroclaw, 50-357, Poland

*Now at Ionospheric and Atmospheric Remote Sensing Group, Jet Propulsion Laboratory, California Institute of Technology, Pasadena, CA, 91109, USA

*Correspondence to*: Natalia Hanna (natalia.hanna@geo.tuwien.ac.at)

**Abstract**. From Global Navigation Satellite Systems (GNSS) signals, accurate and high-frequency atmospheric parameters can be determined in all-weather conditions. GNSS tomography is a technique that takes advantage of these parameters, especially of slant troposphere observations between GNSS receivers and satellites, traces these signals through a 3D grid of voxels and estimates by an inversion process the refractivity of the water vapour content within each voxel. In the last years, the GNSS tomography development focused on numerical methods to stabilize the solution, which has been achieved to a

great extent. Currently, we are facing new challenges and possibilities in the application of GNSS tomography in numerical weather forecasting, the main research objective of this paper. In the first instance, refractivity fields were estimated using two different GNSS tomography models (TUW, WUELS), which cover the area of Central Europe during the period of 29 May - 14 June 2013, when heavy precipitation events were observed. For both models, Slant Wet Delays (SWD) were calculated based on estimates of Zenith Total Delay (ZTD) and horizontal gradients, provided for 88 GNSS sites by

Geodetic Observatory Pecny (GOP). In total, three sets of SWD observations were tested (set0 without compensation for hydrostatic anisotropic effects, set1 with compensation of this effect, set2 cleaned by wet delays outside the inner voxel model), in order to assess the impact of different factors on the tomographic solution. The GNSS tomography outputs have been assimilated into the nested (12- and 36-km horizontal resolution) Weather Research and Forecasting (WRF) model, using its three-dimensional variational data assimilation (WRFDA 3DVar) system, in particular, its radio occultation

observations operator (GPSREF). As only total refractivity is assimilated in GPSREF, it was calculated as the sum of the hydrostatic part derived from the ALADIN-CZ model and the wet part from the GNSS tomography. We compared the results of the GNSS tomography data assimilation to the radiosonde (RS) observations. The validation shows the improvement in the weather forecasting of relative humidity (bias, standard deviation) and temperature (standard deviation) during heavy precipitation events. Future improvements to the assimilation method are also discussed.

## 1 Introduction

Global Navigation Satellite Systems (GNSS) measurements of the microwave signals transmitted between the satellites and the ground-based receivers are affected by the atmosphere (Hofmann-Wellenhof et al., 2001). The signals are bent, attenuated and delayed in two atmospheric layers, namely the ionosphere and the troposphere (Böhm and Schuh, 2013). As the effects caused by both of them can be distinguished, the latter is referred to as the tropospheric delay and stands for the signal delay integrated over the whole ray path (Bevis et al., 1992; Kleijer, 2004; Mendes, 1999). The tropospheric delay can be estimated from differenced GNSS measurements. Therefore, it is usually modelled in a vertical column above each station as Zenith Total Delay (ZTD) – i.e. the observed delay is mapped to the zenith direction and integrated over a certain period of time (Dach et al., 2007). The tropospheric delay is related to the weather conditions in the vicinity of the GNSS station (pressure, temperature, and humidity), therefore it carries valuable meteorological information (Guerova et al., 2016).

In the last years, a huge effort was made to utilize GNSS measurements for operational weather forecasting; as a result, several projects have been conducted to establish processing centres for a continuous estimation of the tropospheric state using GNSS products (Elgered, 2001; Haase et al., 2001; de Haan et al., 2009; Dousa, 2010; Karabatić et al., 2011). Besides from ZTD, another tropospheric parameter, namely an Integrated Water Vapour (IWV) has been derived and assimilated into the Numerical Weather Prediction (NWP) models (Falvey and Beavan, 2002; Gutman et al., 20014). This parameter is calculated based on the wet part of ZTD (i.e. Zenith Wet Delay, ZWD), and is strictly related to the amount of water vapour in the troposphere, thus it is beneficial for meteorological systems since a precise knowledge about humidity in the neural atmosphere is crucial for accurate weather forecasting (Andersson, 2016). Studies have shown that assimilation of GNSS products, either ZTDs or IWVs, into operational NWP models, usually has a positive or neutral impact on the forecast of humidity, rain location, and accumulated rain amount (Poli et al., 2007; Inness and Dorling, 2012; Bennitt and Jupp, 2012). Cucurull et al., 2004; Nakamura et al., 2004; Boniface et al., 2009; Tilev-Tanriover and Kahraman, 2014 demonstrated that the positive impact is most significant during heavy precipitation or storms events.

However, as the tropospheric parameters integrated into a zenith direction reflect horizontal changes in meteorological parameters, they do not provide information about a vertical structure. In order to make better use of information that is contained in the raw GNSS measurements, ZTD can be mapped back into the direction of the satellites in view using mapping functions. Besides, GNSS derived horizontal gradients provide additional information about azimuthal asymmetry, i.e. the horizontal structures in the troposphere (Niell, 1996; Böhm and Schuh, 2004; Böhm et al., 2006). While hydrostatic gradients are caused by differences in tropospheric height as well as regional variations in pressure, wet gradients reflect local variations in the water vapour distribution. In contrast to wet gradients, hydrostatic gradients are usually smaller and show less temporal variations (on average < 0.1 mm / 3 hours vs 0.3 mm / 3 hours for wet gradients, see Ghoddousi-Fard, 2009). In case horizontal gradients are considered in the reconstruction of the GNSS signal delays for each GNSS satellite-receiver pair (along the individual signal path), Slant Total Delays (STDs) are obtained, which represent better the

state of the troposphere, especially in case of strong horizontal humidity gradients during severe weather phenomena in frontal regions (Koch et al., 1997). Operational assimilation of STDs into the MM5 system has shown a positive impact on the representation of water vapour field and precipitation; also a reduction of the spin-up time of the model was noticed (Bauer et al., 2011). An assimilation experiment carried out during a local heavy rainfall event indicated that GPS-STD

assimilation improved rainfall forecast significantly in terms of timing and intensity, when compared to a GPS-ZTD and GPS-IWV assimilation only (Kawabata et al., 2013). However, the issue related to the development of the observation operator for STD is high computational cost of its parts: Forward Operator, Tangent Linear and Adjoint. The Forward Operator requires computing a raytraced delay between the satellite and the ground-based receiver for each STD observation. Another concern is the high nonlinearity of the signal path for low elevation satellites. In the result, the parametrisation for

the Tangent Linear operator is a complex problem and requires a number of iterations. Additionally, the Adjoint operator requires interpolation of the increments to a number of nodes around the signal path. This procedure is computationally expensive, as the signal traverses through the number of voxels in arbitrary direction (not only vertical). Moreover, similar as with IWV and ZTD assimilation, the observation uncertainty is difficult to assign. The STD observations are integrated over the GNSS signal's path, with a different quality of retrieval in the particular parts of the model.

In contrast, if STDs are pre-processed using GNSS tomography principles, the outputs are profiles, similar to the observations for which the operators have been already developed and operationally used in weather models, i.e., Radio Occultation retrievals (Healy, 2007). It was demonstrated in a number of studies that these profiles increase the quality of model fields (Cucurull et al., 2007; Poli et al., 2010; Buontempo et al., 2009; Healy, 2008). The quality of the refractivity profiles is relatively easy to obtain, e.g., by the comparison to the radiosonde profiles, and assigning proper uncertainties to

the coinciding levels (Brenot et al., 2018).

Motivated by the increasing number of GNSS satellites and the build-up of densified ground-based GNSS networks in the 1990s, Flores et al. (2000) proposed a processing method, which allows for reconstruction of structural information from GNSS tropospheric parameters – also known as GNSS tropospheric tomography. This technique is one of the most promising since it uses Slant Wet Delay (SWD) observations together with the principles of tomography to obtain not only

the amount of water vapour, but also its distribution in space and time. For the inversion of the equation system Flores et al. (2000) were using Singular Value Decomposition (SVD) methods together with constraints imposed on the system of equations (smoothing, boundary, and vertical constraints). In the following years, a number of refinements in the technique have been implemented. Inclusion of supplementary wet refractivity data from external data sources into a functional model (Bender et al., 2011; Rohm et al., 2014; Benevides et al., 2015; Möller, 2017) resulted in stabilization of the solution without

need of smoothing constraints, which is especially important during severe weather phenomena with high variability of water vapour in the troposphere. A new approach of parametrization of the domain, using trilinear and spline functions instead of constant values inside each volume element (voxel) showed the substantially smaller maximum error of the solution (Perler et al., 2011; Ding et al., 2018). Another approach, using the Kalman Filter instead of Least-Squares (Rohm et al., 2014), led to better responsiveness to the severe weather. A number of experiments have been carried out showing

positive results for detecting heavy precipitation events (Troller et al., 2006; Rohm and Bosy, 2009; Perler et al., 2011; Rohm et al., 2014; Adavi and Mashhadi-Hossainali, 2015; Chen et al., 2017). Besides the development of technical aspects, which was widely performed in the last years, GNSS tomography should be also tested on its ability to fulfil the requirements of up-to-date weather prediction methods, i.e. on its suitability for assimilation into the NWP models (Innes and Dorling, 2012). The first research on this topic was carried out by Möller et al. (2015). Its aim was to transform wet refractivities derived by GNSS tomography into humidity and temperature profiles, and then assimilate them into the AROME NWP model for a selected test period. Verification was made using surface station measurements, radio sounding data, and precipitation analysis. Tomography data assimilation results compared with the results of ZTD data assimilation, indicated a significantly larger impact of the new technique, especially within 6 to 12 hours of the forecast (a drying effect in the AROME forecast field). As the results clearly showed the potential of 3D refractivity observations, another case study has been performed by Trzcina and Rohm (2018), concerning assimilation of a Near Real-Time (NRT) tomographic solution into WRF model using an operator dedicated to radio occultation (RO) observations of total refractivity (GPSREF; Cucurull et al., 2007). Comparison with several external observations has shown the positive impact within the 6 to 12 hours of the forecast of humidity in autumn, but not in summer. Such assimilation-oriented tomography data analyses are essential for further research on the utilization of GNSS troposphere tomography output as a valuable data source for NWP forecasts.

In this work, we present the assimilation of a 3D wet refractivity field derived by GNSS tomography technique, into the WRF model using the GPSREF (Cucurull et al., 2007) observation operator. The experiment was performed during a heavy precipitation event in Central Europe in order to investigate the impact on the weather forecast. Two tomographic models (TUW, WUELS) and three different approaches of SWD data calculation were used, in order to assess the impact of particular factors on the tomographic solution. Then, the results of several tomographic approaches were discussed in terms of assimilation impact. The structure of the paper is as follows: Section 2 describes the derivation of slant wet delays from GNSS measurements. Section 3 is dedicated to the methodology of GNSS troposphere tomography and provides a detailed description of the models used for tomography processing. Section 4 introduces the methodology of tomographic data assimilation into the WRF model; followed by a description of the meteorological situation under study in Section 5. In sections 6 and 7, the results of tomography and of the assimilation experiments are analysed. The main conclusions are presented in section 8.

## 2 GNSS slant wet delays

The Advanced Global Navigation Satellite Systems Tropospheric Products for monitoring Severe Weather Events and Climate (GNSS4SWEC; http://gnss4swec.knmi.nl/) benchmark campaign has been organized within the EU COST Action ES1206 (Jones et al., 2018). This campaign provided both GNSS tropospheric estimates (with time resolution of 1h for ZTDs and 6h for horizontal gradients) for 88 GNSS sites in Central Europe (see Fig. 2) and data used for validation (ALADIN-CZ data and radiosonde observations). The ZTDs and gradients, which were utilised for the computation of Slant

Wet Delays (SWDs), were estimated in a daily post-processing analysis. For more details about the GNSS processing strategy at GOP, the reader is referred to (Dousa et al., 2016, Section 5.1).

In a first step, the tropospheric parameters for 0, 6, 12, 18 UTC have been selected. Afterwards, for each epoch the Zenith Hydrostatic Delay (ZHD) was computed by means of Saastamoinen model (Saastamoinen, 1972) and removed from the ZTD estimates

$$ZWD = ZTD - ZHD. \tag{1}$$

The required air pressure values have been derived from meso-scale ALADIN-CZ 6-hour forecast data, as provided by the Czech Hydrometeorological Institute (CHMI), and spatially interpolated to each GNSS reference site.

The Zenith Wet Delays (ZWDs) and the horizontal gradients $(G_N, G_E)$ were mapped into direction of the GPS and GLONASS satellites in view above 3° elevation angles. Even though Dousa et al. (2016) have processed only GPS observations, we expect that projection in the direction of all satellites from both constellations will not increase the error of SWDs significantly (Kacmarik et al., 2017). Therefore and for highest consistency with GNSS data processing, the VMF1 mapping function (Böhm et al., 2006) was used for mapping ($mf_w$) of the ZWDs and the Chen and Herring (1997) gradient mapping function ($m_{az}$) was applied for mapping of the horizontal gradient parameters as follows:

$$SWD = ZWD \cdot mf_w(\varepsilon) + G_N \cdot m_{az}(\varepsilon) \cdot \cos(\alpha) + G_E \cdot m_{az}(\varepsilon) \cdot \sin(\alpha) \tag{2}$$

The elevation ($\varepsilon$) and azimuth angles ($\alpha$) of each satellite in view have been computed from broadcast ephemerides. No post-fit residuals were added in the reconstruction of the SWD (Kacmarík et al., 2017). The resulting dataset of SWDs is hereafter referred to as 'set0'.

Based on set0, two additional, more refined datasets were derived, whereby 'set1' compensates for hydrostatic anisotropic effects and 'set2' was cleaned by wet delays 'outside' the inner voxel model (see Table 3 for further details about the voxel model). In both cases, 2D ray-tracing through ALADIN-CZ model level data (6-hour forecast data) was carried for the determination of the hydrostatic delay corrections. For more details about the applied ray-tracer, the reader is referred to Möller (2017).

Within our study period (29 May – June 14, 2013), we retrieved hydrostatic gradients < |0.7 mm|, which corresponds to a signal delay up to |120 mm| at 3 degrees elevation angle. However, under specific conditions, hydrostatic gradients can be as large as +/- 2 mm (see Zus et al. 2019), which corresponds to a signal delay of about ~34 cm (2 mm * 170) at 3 degrees elevation angle. For set1, i.e. for compensation of hydrostatic asymmetric effects, ray-tracing was carried out through ALADIN-CZ 6-hour forecast data. Therefore, ray-traced delays were determined for each GNSS satellite in view, and for equidistant azimuth angles (separated by 30°; Landskron and Böhm, 2018; Zus et al., 2019). From the obtained set of ray-traced hydrostatic delays, the mean hydrostatic delay was computed and removed from the ray-traced hydrostatic delay in direction of the satellite in view. For more details, the reader is referred to Möller (2017), Chapter 6.3.

Figure 1 shows the resulting hydrostatic asymmetric delays, as obtained for all observations within the study period (29 May – 14 June, 2013). In a final step, the hydrostatic asymmetric delays were removed from set0 to obtain set1. As the ALADIN-CZ model pressure error is 0.1 hPa with a standard deviation of about +/-0.6 hPa (Möller, 2017), which corresponds to a hydrostatic delay of about +/-1.5 mm in zenith direction and up to 2 cm at 3° elevation angle, the obtained SWDs (set1) are widely free from hydrostatic effects.

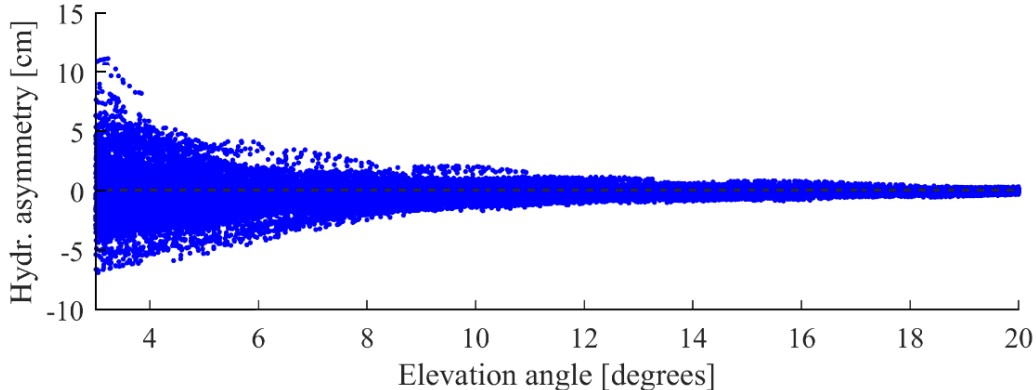

**Figure 1: Hydrostatic component of azimuthal asymmetry, derived from the ALADIN-CZ model level data by ray-tracing.**

Furthermore, for set2 the ray-tracer was applied for determining the slant wet delay outside the inner voxel model (Fig. 2). Therefore, the intersection points between the GNSS signal paths and the boundaries of the inner voxel model were determined, the signal delay outside the inner voxel model was computed by ray-tracing through ALADIN-CZ 6-hour forecast data and finally removed from set0. As a consequence, the outer voxel model is not needed anymore for setting up the tomography model (see Section 3.1 for more details).

## 3 GNSS tomography

For conversion of precise integral measurements (like SWDs) into three-dimensional structures, a technique called tomography has been invented. According to Iyer & Hirahara (1993) the general principle of tomography is described as follows:

$$f_s = \int_S g(s) \cdot ds \qquad (3)$$

where $f_s$ is the projection function, $g(s)$ is the object property function and $ds$ is a small element of the ray path $S$ along which the integration takes place. In order to adapt the tomography principle for GNSS tropospheric delay parameters and to find a solution for Eq. (3), first the troposphere is discretised in volume elements, named voxels, in which the index of

refraction is assumed as constant and the ray path is assumed as straight line. Further, by replacing $f_s$ with $SWD$ and $g(s)$ by wet refractivity ($N_w$), the basic function of GNSS tomography reads:

$$SWD = \sum_{k=1}^{m} N_{w,k} \cdot d_k \tag{4}$$

whereby $N_{w,k}$ is the constant wet refractivity and $d_k$ is the ray length in volume element k. In matrix notation Eq. (4) reads:

$$\underline{SWD} = \underline{A} \cdot \underline{Nw} \tag{5}$$

where $\underline{SWD}$ is the observation vector of size $(n, 1)$, $\underline{Nw}$ is the vector of unknowns of size $(m, 1)$ and $\underline{A}$ is a matrix of size $(n, m)$ which contains the partial derivatives of the slant wet delays with respect to the unknowns. Since Eq. (5) is linear, the partial derivatives of SWD are the ray lengths ($d_k$) in each voxel $k$. A solution for $\underline{Nw}$ is obtained by inversion of matrix $\underline{A}$.

$$\underline{Nw} = \underline{A}^{-1} \cdot \underline{SWD} \tag{6}$$

Unfortunately in GNSS tomography, matrix $\underline{A}$ is usually not of full rank (i.e. has zero singular values). In consequence Eq. (6) is ill-posed, i.e. not uniquely solvable. In order to find a solution for Eq. (6), Truncated Singular Value Decomposition (TSVD) methods are applied (Xu, 1998). Therefore, matrix $\underline{A}$ is split into three components:

$$\underline{A} = \underline{U} \cdot \underline{S} \cdot \underline{V}^T \tag{7}$$

where $\underline{U}$ and $\underline{V}$ are orthogonal matrices and matrix $\underline{S}$ $(n, m)$ is a diagonal matrix. The first $m$ diagonal elements of $\underline{S}$ are the singular values ($s_{k,k}$), all other elements are zero. The ratio between largest and smallest singular value defines the condition number $\kappa(\underline{A})$ of matrix $\underline{A}$.

$$\kappa(\underline{A}) = \frac{|s_{max}|}{|s_{min}|} \tag{8}$$

A condition number near 1 indicates a well-conditioned matrix, a larger condition number indicates an ill-conditioned problem. As a consequence, the tomography solution is sensitive to observation errors, changes in the observation geometry but also to the solving strategy and its parameters defined within the analysis. In the following, the two processing strategies as used for parameter estimation are described more in detail. The general, underlying tomography settings and input data are summarised in Table 3.

## 3.1 Least-squares solution (TUW)

The TUW tomography solution was calculated using the ATom software package (Möller, 2017). It is based on a least-squares adjustment, whereby each observation type is processed individually as partial solution (see Möller, 2017 Eq. 7.46 ff.). This approach allows for proper Truncated Singular Value Decomposition (TSVD), whereby an eigenvalue threshold of 0.006 km² , found by L-curve technique, was set for singular value selection.

Weighting of the GNSS slant observations was based on the elevation angle ($\sin^2 \varepsilon$), with a standard deviation of 2.8 mm for the zenithal observations (ZTD). A height-dependent weighting model was applied to the a priori wet refractivity information, whereby the height-dependent standard deviations were computed by comparison of the a priori data with radiosonde data (stations 10548, 10771) and interpolated to the 15 height levels of the voxel model (see Table 1). In the case, where the altitude of RS station was higher than the lowest tomographic layer (225 m), the values of temperature and specific humidity have been assumed as constant.

The first two TUW solutions are based on SWD set0 and set1. Therefore, the voxel model boundaries were defined according to Table 3 and wet refractivity was estimated in both, the inner and outer domain. In consequence, all available SWDs could be used for estimation of the wet refractivity fields (set0 and set1). The third TUW solution is based on the SWDs of set2. For processing, the outer voxel model was removed since the outer tropospheric delay was already considered in preparation of set2. In consequence, all available SWDs could be used and wet refractivity was estimated for the inner voxel model (set2). Quality control was based on analysis of the SWD residuals ($r$).

$$r_i = SWD_i - A_i \cdot N_w \tag{9}$$

Those residuals larger than 120 times the RMS of the residuals were discarded. The threshold was found empirically and allows for removing large outliers (usually < 2% of SWDs at low elevation angle). The processing stopped after 10 iterations or before, if the change in RMS was lower than 3 percent or if the RMS was lower than 0.5 mm.

## 3.2 Kalman filter solution (WUELS)

Estimation of the wet refractivity distribution in the WUELS solution was performed using TOMO2 software (Rohm and Bosy, 2009, 2011; Rohm, 2012, 2013; Rohm et al., 2014; Zhang et al., 2015), based on a Kalman filter application. In a first step, the state vector of wet refractivity $\underline{N_w}$ and its covariance matrix $\underline{P}$ for epoch $k$ were updated, based on the outputs form a previous epoch ($k-1$):

$$\underline{N_w}_k(-) = \underline{\Phi}_k \underline{N_w}_{k-1}(+) \tag{10}$$

$$\underline{P}_k(-) = \underline{\Phi}_k \underline{P}_{k-1}(+) \underline{\Phi}_k^T + \underline{Q}_{k-1} \tag{11}$$

where $\Phi$ denotes a state transition matrix, indicating predicted changes of the wet refractivity between consecutive epochs; $Q$ is a dynamic disturbance noise matrix. The state transition matrix is a diagonal matrix, set based on a priori information for both epochs where:

$$\underline{\Phi}_{k_{i,i}} = \frac{\left(\underline{N_w}_{apr_k}\right)_{i,1}}{\left(\underline{N_w}_{apr_{k-1}}\right)_{i,1}}. \tag{12}$$

The dynamic disturbance noise matrix was set based on mean wet refractivity changes between epochs, calculated from ALADIN-CZ model data for the whole period. The noise was calculated separately for each height level of the tomographic

domain (see Table 3). After the prediction step, corrections to the state vector and its covariance matrix were computed from the observations. Hereby, the Kalman gain matrix K was defined as follows:

$$\underline{K}_k = \underline{P}_k(-)\underline{A}_k^T\left(\underline{A}_k\underline{P}_k(-)\underline{A}_k^T + \underline{R}_k\right)^{-1} \tag{13}$$

$$\underline{N_w}_k(+) = \underline{N_w}_k(-) + \underline{K}_k\left(\underline{SWD}_k - \underline{A}_k\underline{N_w}_k(-)\right) \tag{14}$$

$$\underline{P}_k(+) = \underline{P}_k(-) - \underline{K}_k\underline{A}_k\underline{P}_k(-) \tag{15}$$

where $\underline{SWD}$ stands for a vector of observations and $\underline{R}$ for its covariance matrix. The vector of observations consists of two parts, the SWD observations and the a priori information derived from ALADIN-CZ model data. Errors of the slant delays $m_{SWD}$ are dependent on the elevation angle $\varepsilon$ and were calculated using the following mapping function:

$$m_{SWD} = \frac{m_{ZWD}}{\sin(\varepsilon)} \tag{16}$$

whereby $m_{ZWD} = 10mm$ is the assumed ZWD error (Kacmarík et al., 2017). Errors of the a priori information were computed by comparison with radiosonde observations (stations 10548, 10771, and 11520), separately for each height of tomographic domain (Table 1). In the case, where the altitude of RS station was higher than the lowest tomographic layer (225 m), the values of temperature and specific humidity have been extrapolated.

For tomography processing, all available SWD observations above 3° elevation angle were taken into account and the signal paths were assumed as straight-lines. The WUELS tomography solutions (set0 and set1) are based on the SWD observations from set0 and set1, respectively. The voxel model boundaries were defined according to Table 3 (both inner and outer voxel model taken into account). A main quality indicator in the WUELS model was based on a comparison of the SWD residuals (see Eq. 9). Observations with residual values $r_i$ larger than three times the standard deviation were removed from the solution (approx. 4% of observations, mainly at low elevation angles). The filter process was stopped after five iterations. Also, 14 stations with a number of removed observations higher than 150 (for the whole period) were rejected.

Table 1: A priori model error ($m$) and dynamic disturbance noise values (Q) as obtained by comparison of the a priori model data with radiosonde data for each level of the TUW and WUELS tomography model.

| | height [m] | 225 | 675 | 1170 | 1715 | 2313 | 2972 | 3697 | 4494 | 5371 | 6336 | 7397 | 8564 | 9848 | 11260 | 12814 |
|---|---|---|---|---|---|---|---|---|---|---|---|---|---|---|---|---|
| TUW | $m$ [ppm] | 4.95 | 4.54 | 5.04 | 5.25 | 4.15 | 5.08 | 3.55 | 2.89 | 2.33 | 1.03 | 0.59 | 0.29 | 0.10 | 0.02 | 0.01 |
| WUELS | $m$ [ppm] | 10.57 | 5.56 | 5.40 | 5.95 | 5.48 | 4.55 | 4.29 | 3.25 | 2.41 | 1.65 | 1.12 | 0.57 | 0.23 | 0.06 | 0.02 |
| WUELS | $Q$ [ppm] | 8.62 | 5.49 | 5.54 | 5.17 | 4.62 | 4.39 | 3.68 | 2.65 | 1.97 | 1.32 | 0.71 | 0.27 | 0.07 | 0.02 | 0.01 |

## 4 WRF Assimilation operator and settings

### 4.1 WRF model

The Weather Research and Forecasting (WRF) model is a mesoscale NWP system designed for both, atmospheric research and operational forecasting needs. It provides 1) two different numerical cores (the Non-hydrostatic Mesoscale Model core (NMM) for operational use and the Advanced Research WRF core (ARW) for research studies), 2) a data assimilation system, 3) a software architecture enabling parallel computation and system extensibility (Schwitalla et al., 2011).

In the WRF model, a nest simulation can be performed. The nest is a finer-resolution model run, which can be embedded simultaneously within a coarser-resolution (parent) model run, or run independently as a separate model forecast. The nested solution allows for the low-resolution model run within the parent domain and the high-resolution model run in the nested domain, what helps enhance the efficiency of the model and reduces the time of calculation of the high-resolution part of the model. Nesting enables also to run the model in a high-resolution very small domain, avoiding a mismatch time and spatial lateral boundary conditions (Skamarock et al., 2008). The coarse grid and the fine grid can interact, depending on the chosen feedback option. Nested grid simulations can be produced using either 1-way nesting or 2-way nesting. In both, the 1-way and 2-way simulation modes, the fine grid boundary conditions (i.e., the lateral boundaries) are interpolated from the coarse grid forecast. In a 1-way nest, the only information exchange between the grids comes from the coarse grid to the fine grid. In the 2-way nest integration, the fine grid solution replaces the coarse grid solution for coarse grid points that lie inside the fine grid. The latest allows information exchange between the grids in both directions (Skamarock et al., 2008).

In this research, the WRF model includes two domains (see Fig. 2): a parent domain (d01), which spans the area of almost the whole Europe, with 36 km horizontal spacing, and a nested domain (d02) within the area of Central Europe, with 12 km horizontal spacing (Fig. 2). The information between the domains flow in both directions (2-way nested run). The vertical resolution of the model includes 35 pressure levels, whereby the model top is defined at 50 hPa air pressure. The model background (initial and boundary conditions) is provided by the National Centers for Environmental Prediction (NCEP) FNL (Final) operational global analysis data with 1° x 1° horizontal resolution and 26 vertical layers. These data are available at http://rda.ucar.edu/datasets/ds083.2/.

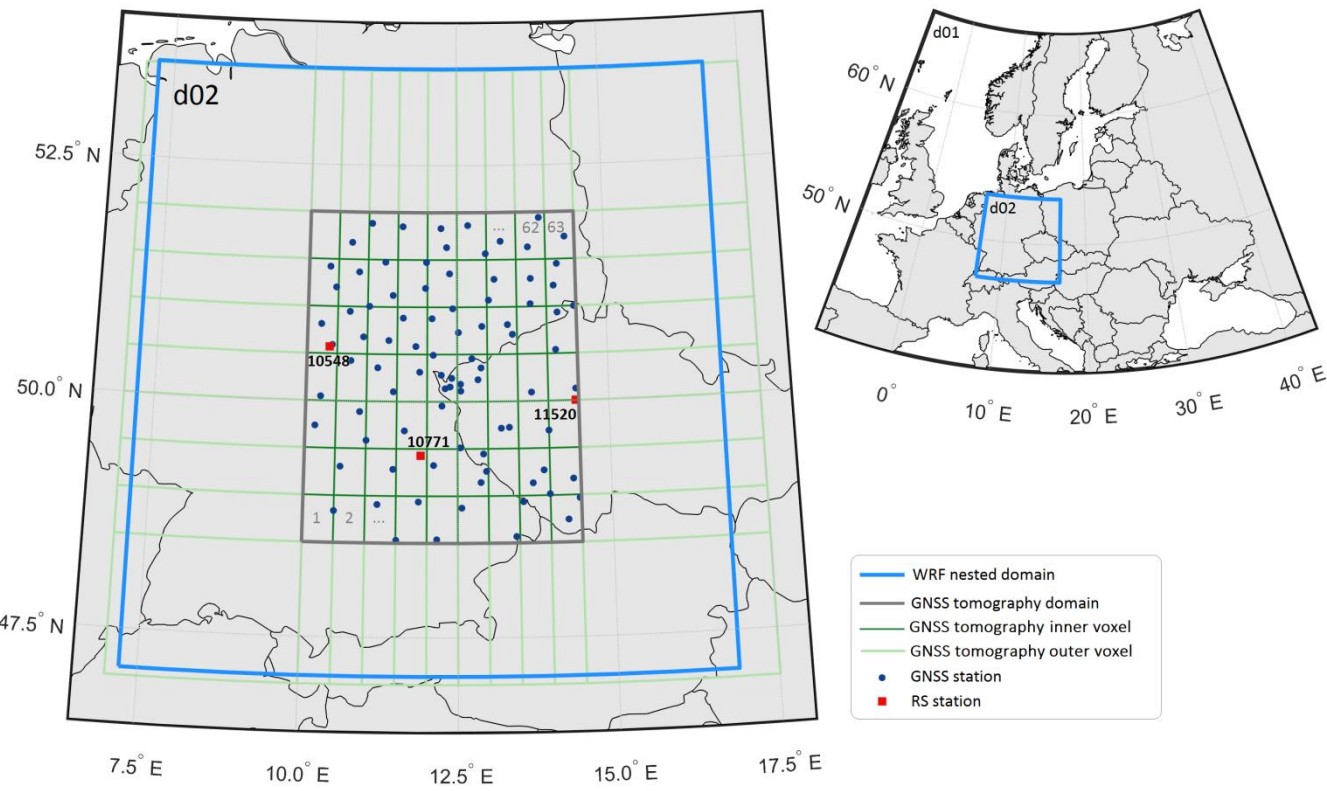

**Figure 2: The geographical extensions of the WRF domains used in this research (d01 = coarse grid with 36 km horizontal spacing, d02 = fine grid with 12 km horizontal spacing), GNSS tomography domain (grey line), together with inner (dark green line) and outer (light green line) voxels, including GNSS sites (dark blue dots) and radiosonde stations (red squares).**

## 4.2 Assimilation operator

Data assimilation is the technique by which observations are combined with an NWP product (the first guess or background forecast) and their respective error statistics to provide an improved estimate (the analysis) of the atmospheric (or oceanic, Jovian, etc.) state. Variational (Var) data assimilation achieves this through the iterative minimization of a prescribed cost (or penalty) function. The aim of a variational data assimilation scheme is to find the best least-square fit between an analysis field $x$ and observations $y$ with an iterative minimization of a cost function $J(x)$:

$$J(x) = \frac{1}{2}(x - x_b)^T B^{-1}(x - x_b) + \frac{1}{2}(y - H(x))^T R^{-1}(y - H(x)). \tag{17}$$

In Eq. 17, $x$ is a vector of analysis, $x_b$ is the forecast or background vector (first guess), $y$ is an observation vector, $B$ is the background error covariance matrix, $H$ is an observation operator, and $R$ is the observation covariance matrix. The observation operator $H$, which can be non-linear, converts model variables to observation space. Differences between the analysis and observations/first guess are penalized (damped) according to their perceived error.

In order to assimilate observations into the WRF data assimilation system, a special function, i.e. data assimilation operator ($H$), should be used. Although the WRF DA system does not provide a direct assimilation operator for the GNSS tomography wet refractivity fields, it is possible to apply a radio occultation operator (GPSREF) for the GNSS tomography output data. The GPSREF assimilation operator is used to map model variables to refractivity space (Cucurull et al., 2007):

$$N = N_h + N_w = 77.6\left(\frac{P}{T}\right) + 3.73 \times 10^5 \left(\frac{P_v}{T^2}\right), \tag{18}$$

where $P$ is the total atmospheric pressure (hPa), $T$ is the atmospheric temperature (K), and $P_v$ is the partial pressure of water vapor (hPa). The GPSREF operator enables assimilation of total refractivity ($N$) at each observation height. Since GNSS tomography provides $N_w$ only, the hydrostatic (dry) component of refractivity ($N_h$) has to be modeled. Therefore we use meteorological parameters (total atmospheric pressure and atmospheric temperature) provided by ALADIN-CZ model.

The ALADIN-CZ model is a local area, hydrostatic model provided by Czech Hydrometeorological Institute (CHMI). Model simulations run at 4.7 km x 4.7 km horizontal resolution and 87 hybrid-model levels. NWM 3D analysis fields are provided in GRIB format for 00, 06, 12, and 18 UTC together with 6-hours forecast fields with one hour resolution (Kacmarik et al, 2017). In this study, the analysis data of the ALADIN-CZ model is used to: 1) Compute pressure values at the height of GNSS station to reduce the hydrostatic part from the Slant Total Delays (STDs); 2) Compute temperature and pressure at the height of each $N_{wet}$ observation in order to model the hydrostatic part of refractivity. In order to calculate the hydrostatic component of refractivity, the meteorological parameters provided by ALADIN-CZ are interpolated to each observation height. Finally, total refractivity field, i.e. the sum of $N_h$ and $N_w$, is assimilated.

### 4.3 Assimilation settings

The 3DVar assimilation of five sets of tomography data (TUW set0, TUW set 1, TUW set2, WUELS set0, WUELS, set1) and the control run of the base weather forecast (without assimilation) has been performed. In order to allow the NWP model to approach its own climatology after being started from the initial conditions (NCEP final data), the assimilation was performed six hours after model run (spin-up time). The analysis has been performed at 00, 06, 12, and 18 UTC. Each analysis includes 18 hours forecast with one hour resolution. Table 2 shows the data assimilation settings. In the process of assimilation, only the data from the voxels crossed by the GNSS signal were used.

**Table 2: Applied data assimilation settings.**

| Horizontal resolution | parent domain: 36 km x 36 km |
|---|---|
| | nested domain: 12 km x 12 km |
| Vertical layers | 35 |
| Method | 3DVar |
| | radio occultation observation operator GPSREF |
| Initial and boundary | NCEP FNL 1° x 1° |

| conditions | |
|---|---|
| **Assimilation window** | 1 h |
| **Model run** | 00, 06, 12, 18 UTC<br>6 hours of model integration + 18 hours of forecast lead time |
| **Physic options** | Yonsei University (YSU) Scheme (Hong et al., 2006) for the Atmospheric Boundary Layer (ABL) parametrisation;<br>Revised MM5 scheme (Jimenez et al., 2012) for surface layer;<br>Unified Noah land surface model (Tewari et al., 2004);<br>Dudhia scheme for shortwave radiation (Dudhia, 1989);<br>Kain-Fritch scheme for the cumulus parametrization (Kain, 2004) |

## 5 Case study

Our studies are based on the complex dataset collected within the European COST Action ES1206, as described in detail by Dousa et al. (2016). The study area comprises central parts of Europe (Fig. 2). The study period (29 May to 14 June, 2013) covers the events of heavy precipitation, which finally led to severe river flooding in all catchments on the Alpine north-side (Switzerland, Austria, and southern Germany) and of the mountain ranges in Southern and Eastern Germany as well as in the Czech Republic.

As reported by Grams et al. (2014), the core period of the heavy precipitation occurred from 31 May 2013, 00:00 UTC to 3 June 2013, 00:00 UTC. Within these 72 hours, accumulated precipitation of 50 mm was reported in regions of Eastern Switzerland, the Austrian Alps and Czech Republic, and exceeded even 100 mm in several regions of the Northern Alps. The events of heavy precipitation are connected to the presence of three cyclones: "Dominik", "Frederik", and "Günther". These cyclones formed over (South-) Eastern Europe as a "cluster" with very similar tracks, and were rather shallow systems with relatively high values of minimum sea level pressure. These systems unusually track westward, maintaining a northerly flow against the west-east oriented mountain ranges in Central Europe. However, within this period their movement was blocked by an anticyclone located over the North Atlantic, which forced the cyclones to form an equatorward conveyor belt.

The GNSS tomography solutions cover the region of East Germany and western parts of Czech Republic, including the Erzgebirge (see Fig. 2). Within this area, the equatorward flowing warm air masses started to lift up, which goes along with a local maximum of precipitation of more than 75 mm within the core period of heavy precipitation. For tomography processing, the study area was divided into an inner and an outer voxel model. The parameters of the tomography model and model settings are summarized in Table 3. From the benchmark dataset, 88 GNSS sites could be identified within the inner model, with an average distance of 48 km and altitude ranging from 70 m to 885 m.

**Table 3: Applied tomography model settings.**

| | |
|---|---|
| **Period** | 29[th] of May – 14[th] of June 2013 |
| **GNSS stations network** | number of GNSS sites: 88<br>average distance: 48 km<br>altitude of stations: 70 m – 885 m |
| **Tomography domain** | latitude: 48.5° – 52.0° (0.5° resolution, approx. 55 km)<br>longitude: 10.0°– 14.5° (0.5° resolution, approx. 43 km)<br>height: 225 – 12814 m<br>outer voxel model: latitude +/- 1.5° (approx. 170 km), longitude +/-3° (approx. 260 km) |
| **A priori data** | ALADIN-CZ 6-hour forecast:<br>horizontal resolution: 4.7 x 4.7 km<br>vertical layers: 87 model levels<br>time of analysis: 00:00, 06:00, 12:00, 18:00 UTC<br>forecast ranges: 0, 1, 2, 3, 4, 5, 6 h<br>coordinates: non-rotated Lambert projection according to CHMI specification |
| **SWD observations** | set 0: SWDs (GPS + GLONASS, ZHD from ALADIN-CZ)<br>set 1: SWDs (GPS + GLONASS, ZHD from ALADIN-CZ, hydrostatic gradients removed)<br>set 2: SWDs (GPS + GLONASS, ZHD from ALADIN-CZ, outer delay removed) |
| **Cut-off angle** | 3° elevation angle |
| **Time settings** | every 6 hours (00, 06, 12, 18 UTC) |
| **Quality indicators** | TUW: outlier tests based on the post-fit residual RMS<br>WUELS: filtration of dSWD based on synthetic data for detecting outlier observations |

## 6 Tomography results

Based on the analyses carried out by Möller (2017), it is expected that the quality of the tomography wet refractivity
5  solution differs significantly with model altitude. Thus, in the following the five TUW and WUELS tomography solutions
are validated against radiosonde measurements. Besides, intra-technique comparisons highlight the impact of the
tomography settings on the obtained wet refractivity fields.

### 6.1 Intra-technique comparison

Over the study period of 17 days, in total 68 tomography epochs (four solutions per day) have been processed for
10  each tomography solutions (TUW set0, set1, set2 and WUELS set0 and set1). Based on the wet refractivity differences
between TUW set0 and set1, bias and standard deviation were computed for each voxel. Figure 3 shows the results for the 63
(horizontal) x 15 (vertical) voxels of the inner voxel model, whereby voxel number 1 indicates the lower South-West corner
and the voxel number 63 the North-East corner of the domain. The comparison between the TUW set0 and TUW set1 shows

that the impact of anisotropy correction on the tomography wet refractivity field is very small (up to 1 ppm for bias and standard deviation). For the WUELS model, the differences reach maximally 2 ppm in terms of bias. In general, in the case of standard deviation, the differences are up to 1 ppm (not shown).

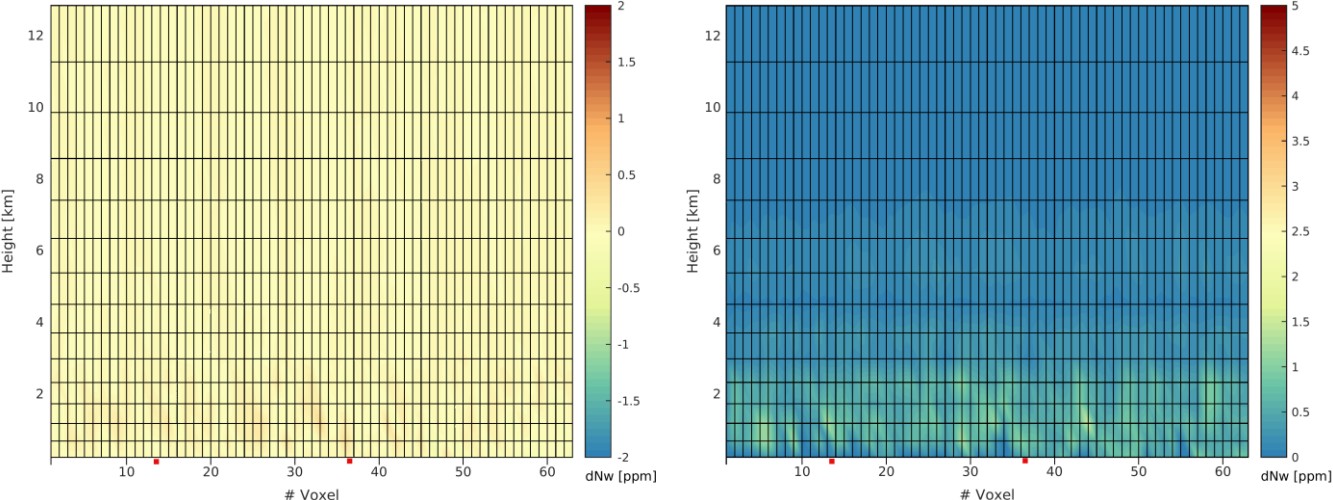

**Figure 3: Bias (left) and standard deviation (right) of the differences in wet refractivity [ppm] between TUW set0 and TUW set1. Analysed period: 29 May – 14 June 2013 (68 epochs). Red squares denote location of the RS stations.**

A similar analysis has been carried out for TUW set0 and TUW set2 (Fig. 4). The major differences between both solutions are caused by the different approaches for compensation of the outer voxel delay. While in set0 the tropospheric delay in the outer voxel model was estimated, in set2 the outer delay was removed beforehand by ray-tracing through
ALADIN-CZ 6-hour forecast data. Largest offsets between both TUW solutions appear in Northern parts of the study area (voxels columns #55-63). In this part, the tropospheric delay is systematically overestimated (compared to ray-traced delays) in the outer voxel model and in consequence, underestimated in the inner voxel model. This leads to the positive bias as visible in Fig. 4. In all other parts of the voxel model, the differences are widely averaged out over the study period of 17 days. However, the standard deviation shows that variations over time cannot be avoided. Especially between the 31st of May
18 UTC and the 4th of June 18 UTC, larger standard deviations were detected (up to 10 ppm; not shown). These variations are caused by changes in the observation geometry but also by changes in atmospheric conditions not described by the forecast data (error source in set2).

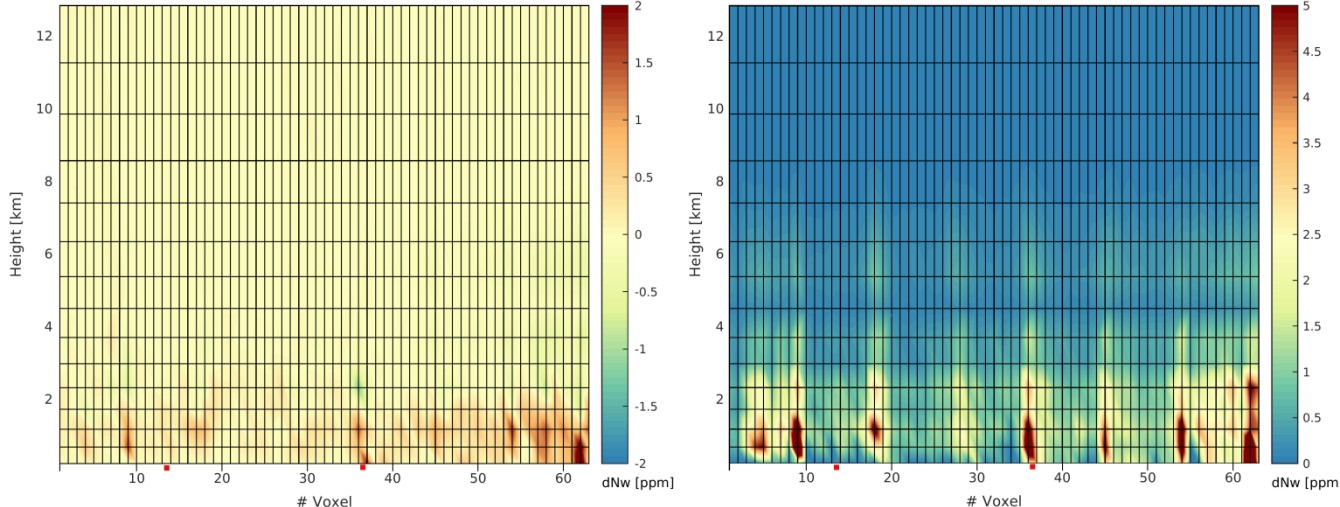

**Figure 4: Bias (left) and standard deviation (right) of the differences in wet refractivity [ppm] between TUW set0 and TUW set2. Analysed period: 29.05. – 14.06.2015 (68 epochs). Red squares denote location of the RS stations.**

Figure 5 presents the differences for bias and standard deviation between both TUW set1 and WUELS set1. Since both tomography solutions are based on same input data (set1), similar wet refractivity fields are obtained. The largest discrepancies are visible within the boundary layer, i.e. the lowest 1-2 km of the atmosphere, and for the layer of 3-6 km height. The TUW solution tends to produce larger wet refractivity fields in specific voxels near surface (#19, 20, 24, 25, 35,

10  38, 41, 48, 50,…), in return slightly lower values in the 2-3 layers above, and then slightly higher values for the heights of 3-6 km (up to 5 ppm). Above the 6 km altitude, the absolute bias and the standard deviation between TUW and WUELS solutions decrease significantly (below 3 ppm). The discrepancies between both models are caused by different approaches of the tomography settings. The tomography models differ in application of quality indicators. In the WUELS approach, the GNSS stations that frequently failed quality-check, are removed from the solution; whereas in the TUW solution all GNSS

15  stations are taken into account in every epoch. As a result, 14 GNSS stations were rejected from the WUELS model but not from the TUW model. Locations of these stations within the voxel domain are presented in Fig. 5 (grey triangles). In most cases, the locations of the rejected stations correspond with the largest discrepancies between the models, e.g. voxels number 19, 37-40, 45-49. Also, different weightings of the a priori data were applied in both solutions which results in noticeable discrepancies between the outputs of the models.

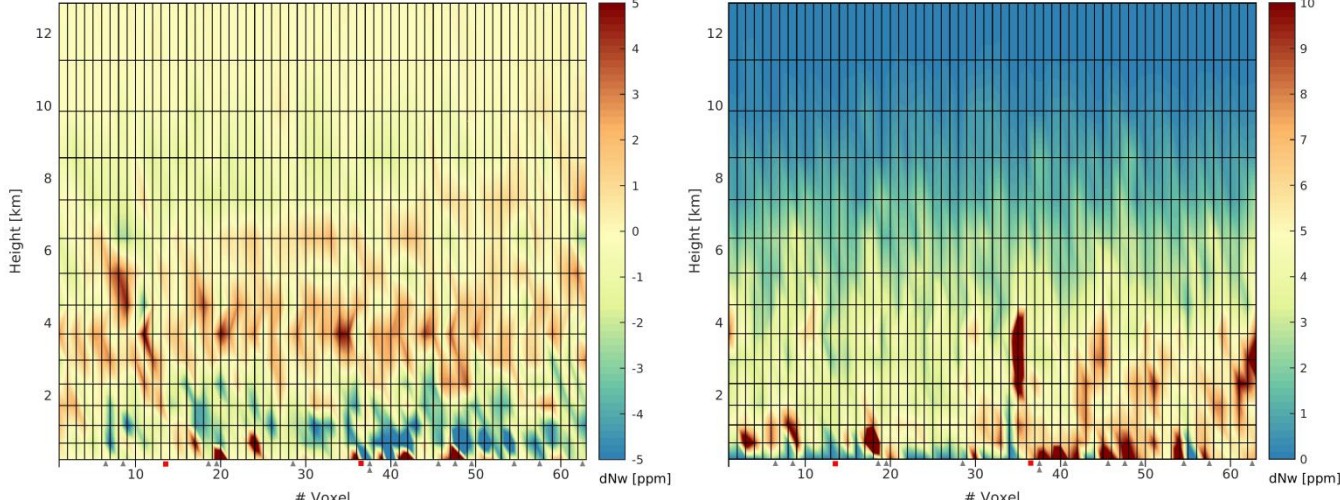

**Figure 5: Bias (left) and standard deviation (right) of the differences in wet refractivity [ppm] between TUW set1 and WUELS set1. Analysed period: 29 May – 14 June 2013 (68 epochs). Grey triangles denote GNSS stations removed from the specific voxels of the WUELS solution (one triangle stands for one rejected GNSS station), red squares denote location of the RS stations.**

Based on our analysis, we conclude that the removal of the hydrostatic anisotropic effects does not influence significantly the tomographic outputs (approx. 1 ppm in standard deviation). The approach applied for the removal of the outer parts of the GNSS signal (i.e. set0 and set2) has a larger impact on the results, however, not higher than 5 ppm in terms of the standard deviation. The presented analysis shows that the greatest impact on the GNSS tomography results comes from the applied model (TUW set1 and WUELS set1), where the standard deviation is up to 10 ppm.

## 6.2 Time series of integrated zenith wet delays

From the obtained wet refractivity fields, time series of ZWDs were computed for the 88 GNSS sites within the study area by vertical integration using Eq. 19,

$$ZWD = 10^{-6} \sum_{H_0}^{H_t} N_w \cdot dh \tag{19}$$

where $H_0$ is the height of the GNSS site and $H_t$ is the height of the voxel top (at about 13.5 km height above mean sea level). Beforehand, $N_w$ was horizontally linearly interpolated from adjacent voxel centre points to GNSS site and the vertical resolution was further increased to 20 m. Figure 6 shows the derived ZWD time series for the entire study period of 17 days with 6 hour temporal resolution, exemplary for GNSS site WTZR.

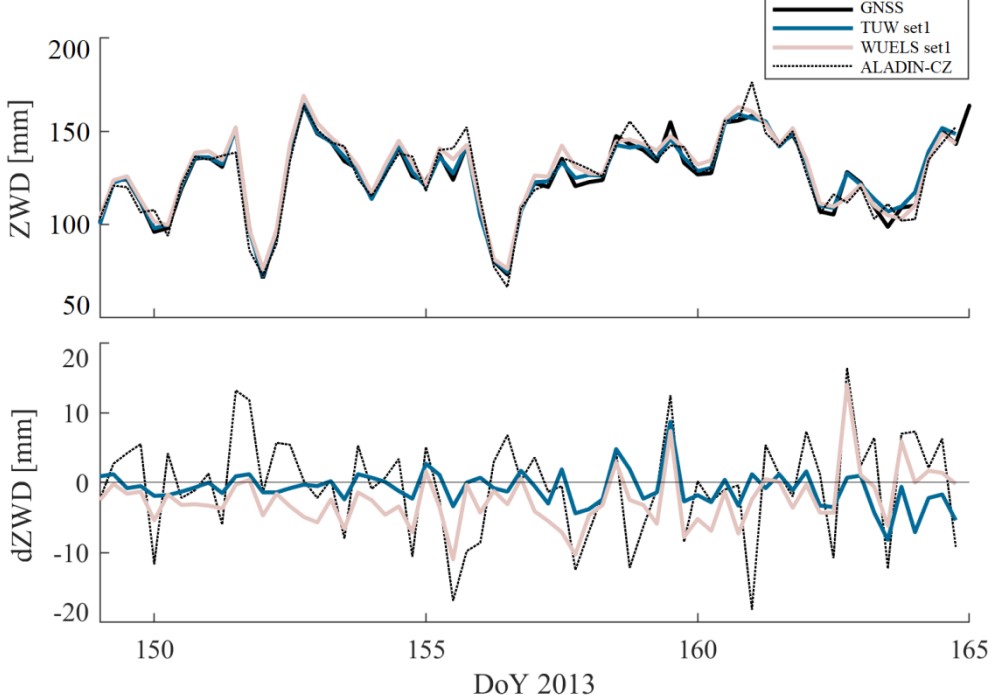

**Figure 6: GNSS-derived (black) and tomography-derived (blue and red) ZWD time series at GNSS site WTZR (Wettzell, Germany). The dashed line shows the ZWD values derived from the ALADIN-CZ model. The top plot shows the absolute values and the bottom plot highlights the ZWD differences with respect to the GNSS-derived ZWDs.**

Both tomography solutions are more consistent with the GNSS-derived ZWDs than the ALADIN-CZ data. The ZWDs derived from ALADIN-CZ 6-hour forecasts are occasionally biased, a few hours 'ahead' or 'behind' the GNSS-derived ZWDs. The WUELS solution is negatively biased, while discrepancies between GNSS and TUW are smaller and the systematic error is reduced. Table 4 shows the statistic of the ZWD differences for GNSS site WTZR but also the mean values over all 88 GNSS sites within the study domain. For both tomography models, the bias in ZWD is larger than for the a priori model (0.6 mm for TUW and -1.8 mm for WUELS), but the standard deviation is reduced (by 6.3 mm for TUW and by 4.1 mm for WUELS).

**Table 4: Statistics of the differences in ZWD for GNSS site WTZR and all 88 GNSS sites within the study domain. The reference solution (GNSS) was obtained by parameter estimation using GNSS phase measurements. The ALADIN-CZ, the TUW set1 and WUELS set1 were computed by vertical integration through the wet refractivity fields: Analysed period: 29th of May until 14th of June 2013.**

| | GNSS minus ALADIN-CZ | | GNSS minus TUW | | GNSS minus WUELS | |
|---|---|---|---|---|---|---|
| | bias(dZWD) | stddev(dZWD) | bias(dZWD) | stddev(dZWD) | bias(dZWD) | stddev(dZWD) |

| | | | | | | |
|---|---|---|---|---|---|---|
| WTZR | -0.5 mm | 7.2 mm | -0.9 mm | 2.6 mm | -2.5 mm | 3.9 mm |
| ALL (88) | 0.1 mm | 9.1 mm | 0.6 mm | 2.8 mm | -1.8 mm | 5.0 mm |

With respect to ZWD, both tomography solutions are closer to the GNSS solution than the a priori ALADIN-CZ 6-hour forecast. Especially the standard deviation of the ZWD differences can be reduced if GNSS tomography is applied (Fig. 7).

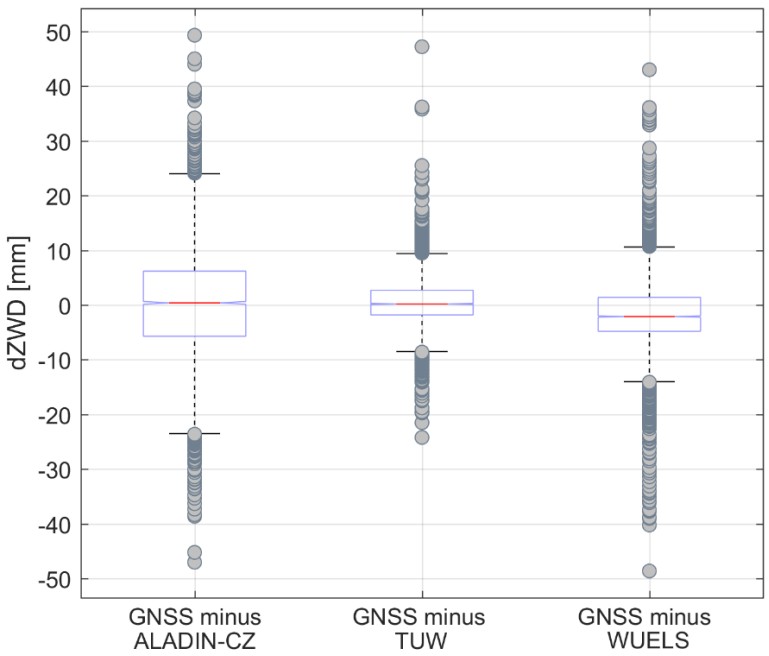

**Figure 7: Box plots of the ZWD differences for ALADIN-CZ, TUW, and WUELS models, compared to the GNSS data for all 88 sites.**

Figure 7 presents the box plots of the ZWD differences for all observations. The number of outliers for ALADIN-CZ model (141) is lower than for the tomography solutions (275 for TUW and 314 for WUELS). This is a result of the higher value of standard deviation for ALADIN-CZ model than for the tomographic models. In addition, significant differences are visible between both tomography solutions. The WUELS solution tends to overestimate the water vapour content in the atmosphere slightly, this goes along with a higher variability of the ZWD differences (about 2 times larger standard deviation than for the TUW solution). This shows that weighting of a priori data and applied quality control methods have a great impact on the results of tomography solutions.

### 6.3 Validation of tomography results with radiosonde data

Radiosonde measurements of temperature ( $T$ ) and dew point temperature ( $T_d$ ) were downloaded from (https://ruc.noaa.gov/raobs/) for station 10548 near Meiningen, Germany (Lat: 50.57°, Lon: 10.37°, H=450m within voxel

column #37) and 10771 near Kümmersbruck, Germany (Lat: 49.43°, Lon: 11.90°, H=420m within voxel column #13). The temporal resolution of the measurements is 12 hours (10548) and 6 hours (10771), respectively. Dew point temperature and temperature were converted into water vapour $e$ (Sonntag, 1990)

$$e = 6.112 \cdot e^{\frac{17.62 \cdot T_d}{243.12 + T_d}} \tag{20}$$

and wet refractivity

$$N_w = k_2' \cdot \frac{e}{T} + k_3 \cdot \frac{e}{T^2} \tag{21}$$

with $k_2' = 22.9744 \, K/hPa$ and $k_3 = 375463 K^2/hPa$, see Rüeger, 2002, best average. For comparison, the obtained profiles of wet refractivity ($N_w$) were linearly interpolated to the voxel centre heights (see Table 1).

Figure 8 shows wet refractivity profiles as obtained at radiosonde site RS10548 on the 6[th] of June 2013, 00 UTC
and radiosonde site RS10771 on the 13[th] of June 2013, 00 UTC. Both plots were selected to highlight typical characteristics of the tomography solution. Due to proper weighting, the tomography solution can correct deficits in the a priori model (see the lower 2 km of TUW set1 in Fig. 8 left and the WUELS set1between 2 and 5 km height in Fig. 8 right). Sharp changes in wet refractivity can be recovered from TUW tomography solution in the boundary layers and from the WUELS solution in the mid-troposphere (related to weighting model). The tomography solutions are still affected by reconstruction errors
(artefacts in the Nw profiles as visible in Fig. 8 left between 2 and 5 km height).

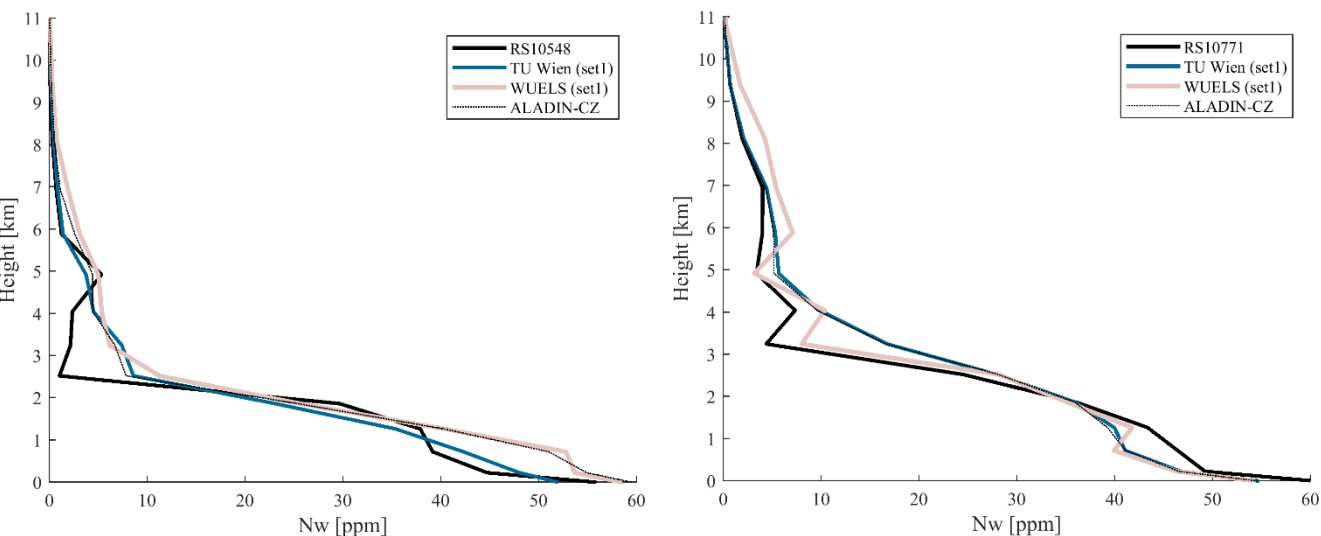

**Figure 8: Wet refractivity profiles derived from radiosonde launches, ALADIN-CZ 6-hour forecast data, TUW and WUELS tomography set1 for the 6[th] of June 2013, 12 UTC (left) and the 13[th] of June 2013, 00 UTC (right), respectively.**

Figure 9 shows the corresponding statistics (bias, standard deviation, and RMS) over all analysed epochs (34 for
RS10548 and 68 for RS10771), separately for each radiosonde site. While RS10548 is located nearby a GNSS site, RS10771

lies within a voxel in which no GNSS site is located. Nonetheless, the quality of the tomography solution does not vary significantly according to the locations of RS stations. Up to 2 km height, the statistics for both tomographic models are on the same level. In the upper parts of the troposphere, the RS - TUW comparison shows similar accuracy as the RS - ALADIN-CZ data, whereas the standard deviation and RMS of the WUELS model is noticeably higher.

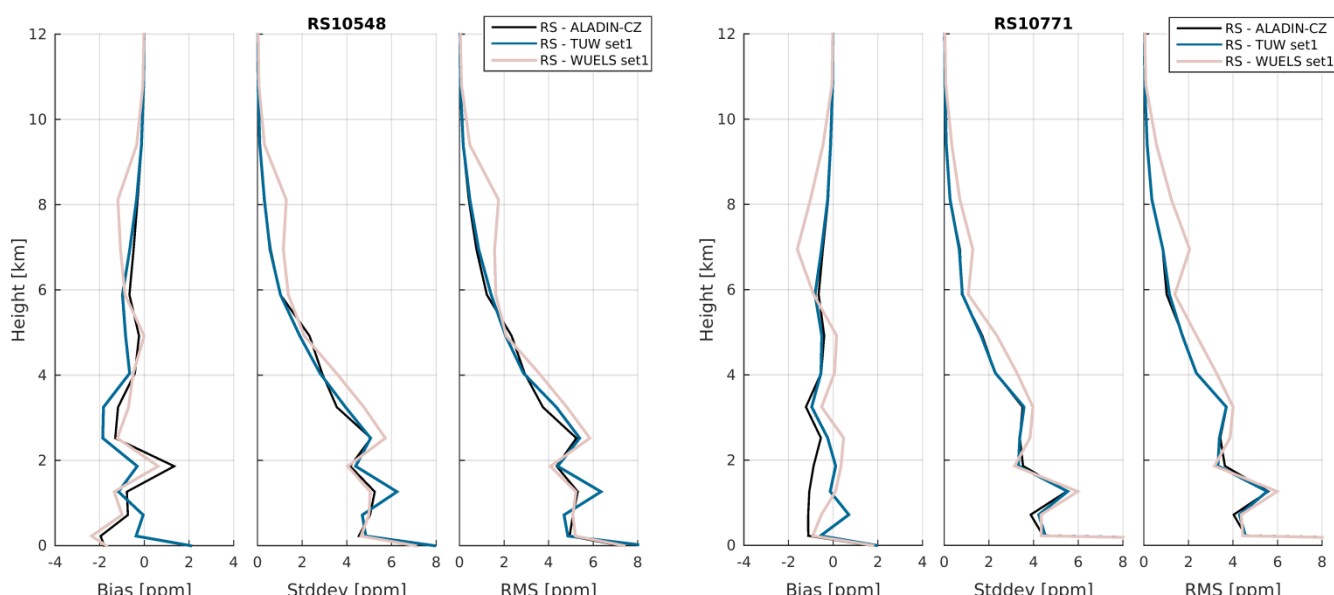

**Figure 9: Statistics of the differences in wet refractivity at radiosonde site RS10548 (left) and RS10771 (right).**

Figure 10 shows the differences between TUW set0 and TUW set2, separately for each radiosonde site. In fact, the strategy for compensation of the outer delay has only a small impact on the tomography solution. Thereby, the impact is
10   independent from the location of the radiosonde site within the inner voxel model but mostly related to the tomography settings. The largest differences are visible in the lower 2 km of the atmosphere. However, the overall impression is that set0 (estimation of the tropospheric delay in the outer voxel model) provides slightly better results than set2 (compensation of the outer delay by ray-tracing through ALADIN-CZ 6 hour forecast data). This is most likely related to the quality of the weather model forecast data during the period of extreme precipitation. Nevertheless, since both solutions are rather close to
15   each other, especially with respect to standard deviation, only small improvements are expected by ray-tracing through more reliable weather forecast data.

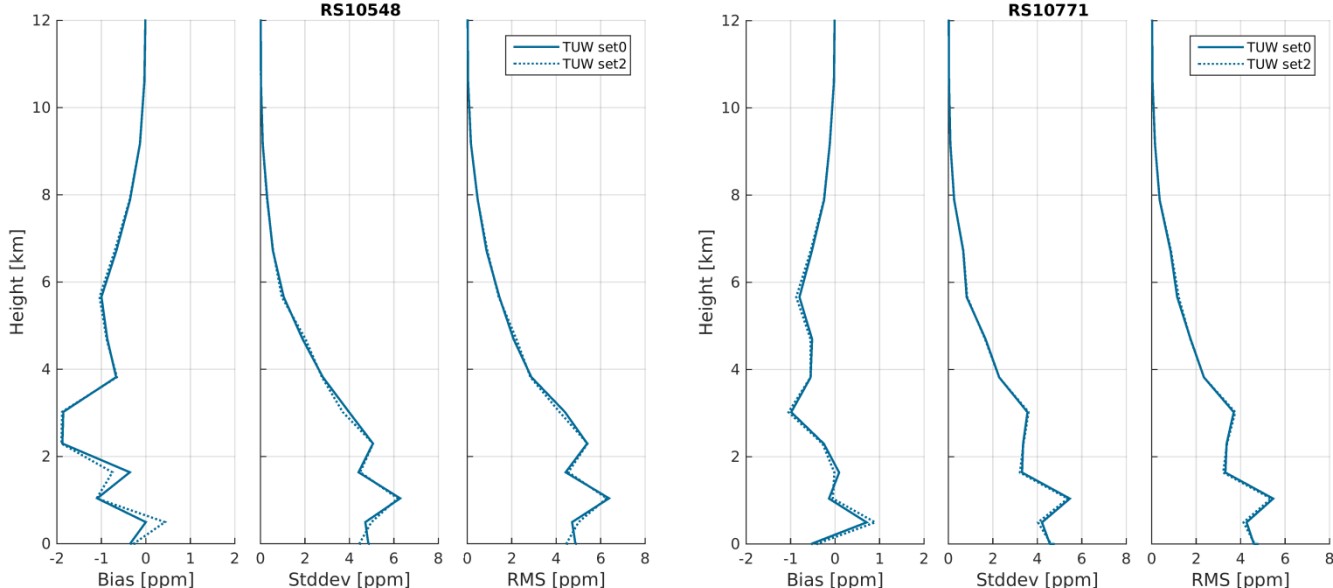

**Figure 10: Statistics of the differences in wet refractivity between radiosonde data and various tomography solutions. Differences between set0 and set2 are related to the strategy for compensation of the outer tropospheric delay.**

## 7 Assimilation results

### 7.1. Diagnosis output

The WRFDA system and the GPSREF operator are equipped with a quality control diagnostic tool, which allows for verification of all input data before assimilation. As a result, not all of the refractivity observations are actually assimilated into the model. Figure 11 presents a percentage of successfully assimilated observations for each tomography set, as a function of height. The height range with the largest number of assimilated observations is between 4 and 10 km. For these heights, more than 90% observations of the TUW solutions and about 50-80% of those from the WUELS solution were assimilated. Below 4 km, the percentage of assimilated observations grows systematically with height, from 0% at surface to 70% for TUW and 40% for WUELS. Above 10 km, no observations were assimilated, as they were removed in the quality control process.

Since the comparison of tomographic observations with radiosonde data showed that in general the TUW solutions have smaller errors than the WUELS solutions, the number of observations that passed the quality control is, in general, connected to the quality of the tomographic data. Because of the restrictive quality control process in the GPSREF operator, some exception from this rule can be noticed in the lower (0-4 km) and the upper (10-12 km) troposphere, where almost all observations have been eliminated from the assimilation. The radio occultation observations, to which the GPSREF operator is dedicated, very rarely reach the lowest level of the troposphere, whereas they are very accurate in the upper level.

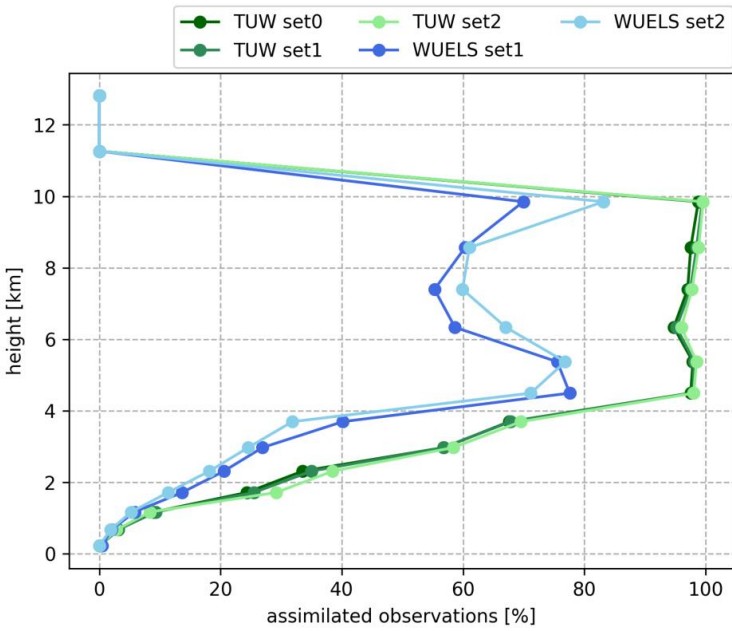

**Figure 11: Percentage of observations successfully assimilated for TUW (green) and WUELS (blue) models in a function of height.**

Apart from the number of successfully assimilated observations, the reason of the rejection was also studied. In the quality control diagnostics of the GPSREF operator, each observation is assigned to one group, based on the information of acceptance or rejection (and its cause). Figure 12 presents the results of the quality control check, separately for each group and for each tomography solution as function of height in colour lines. For both tomography solutions, a large number of observations is assigned to a group 0 (orange line) that denotes successfully assimilated data. The number of observations from this group corresponds well with the one already presented in Fig. 11. The quality flags with numbers -88, -77, and -3 are the general flags of the WRFDA system, whereas the numbers from -31 to -36 are the flags assigned by the GPSREF operator.

The general flags with numbers -88, -77, and -3 denote respectively: data below the model's terrain, data laying outside of horizontal domain, data failing maximum error check. Observations assigned to the first group (blue line) occur in the surface layer only, in number of about 2000 observations for the TUW model and 1200 for the WUELS. The second group (yellow line) includes observations from the two highest layers (11.260 km and 12.814 km). The number of observations assigned to the third group (green line) is about 0 at all heights, only for the WUELS model at heights 4 – 8 km the number slightly grows up to about 50 observations.

The quality flags assigned by the GPSREF operator (from -31 to -36) are connected with the values of assimilated refractivity data. The first type of diagnostics is based on the comparison between the assimilated observations and the background values of refractivity (Cucurull et al. 2007). The discrepancies between the two refractivity values should not be larger than 5% (below 7 km) or 4% (7 – 25 km) of the mean value. Observations that do not meet this requirement are

assigned to the group -31 (below 7 km, red line) or -32 (above 7 km, purple line). Additionally, if an observation gets the flag -31, all observations in the same vertical profile (same latitude and longitude) below that observation, are also assigned to the flag -31 and they are not assimilated into the model. For the TUW model, the largest number of observations with flag -31 is at height of 0.675 km (about 3000 observations); this number systematically decreases with height, to about 0

observations at 4 km. For the WUELS model, the largest discrepancies between the observations and the background occur between the heights of 0.675 km and 2.972 km (2000 – 3000 observations); no significant discrepancies are noticed above 6 km. The second type of quality check inside the GPSREF operator is based on a refractivity lapse rate (Poli et al., 2009). The -34 flag (brown line) is assigned to the observations where $\frac{dN}{dz}$ is smaller than -50 km$^{-1}$, whereas the -35 flag (pink line) indicates observations where an absolute value of $\frac{d^2N}{dz^2}$ is larger than 100 km$^{-2}$. For the TUW model, there are no observations

rejected from the assimilation based on the lapse rate of refractivity. It indicates internal coherency of the model's output. In the WUELS model, for each layer above 6 km, more than 1000 observations were rejected from the assimilation process based on the refractivity lapse rate. The last type of quality check (flag -36, grey line) is based on the discrepancies between observations and background, as proposed by Cucurull (2010). In the TUW model, these flags occur mainly in the bottom parts of the troposphere (0-4 km), whereas in the WUELS model inconsistency between model and background is noticed

also for the higher parts (5 – 8 km). The results of the quality control process show, in general, that the number of assimilated data for each model is related to the quality of the observations. However, the exception from this rule can be noticed in the lowest (0-2 km) and the highest (above 10 km) parts of the troposphere, where the number of rejected observations seems to be too high.

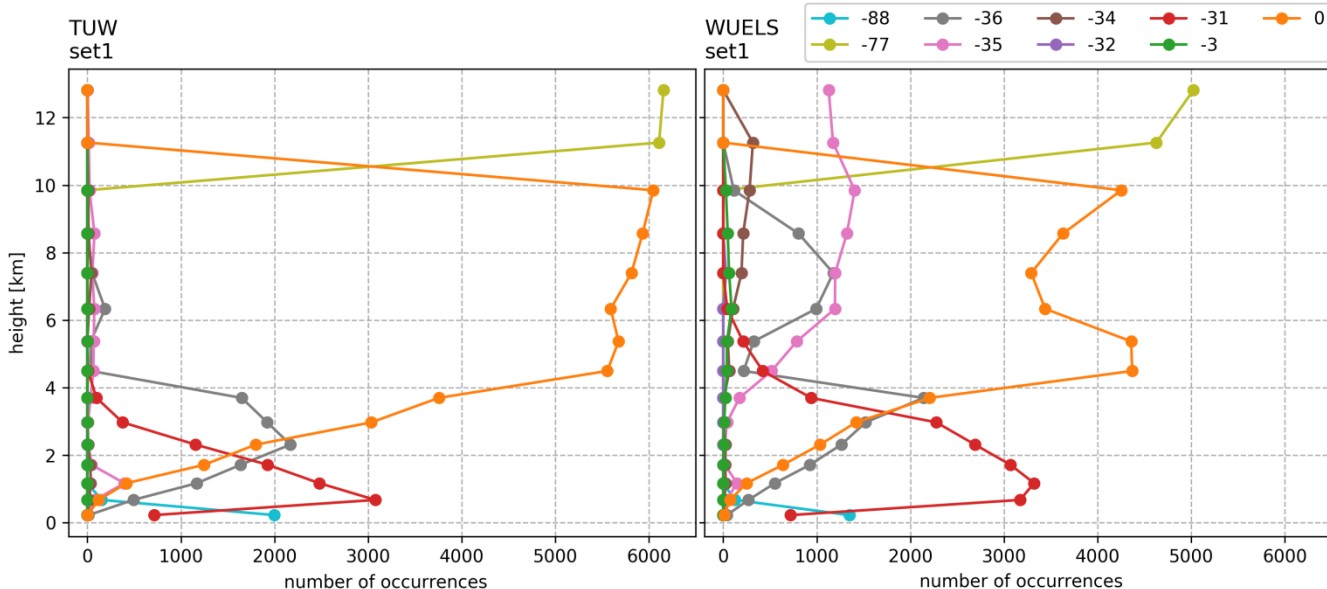

**Figure 12: Number of flagged observations as function of height (flags explanation in the text) - separately for TUW set1 (left) and WUELS set1 (right).**

### 7.2. Assimilation output results compared to base run

5        In order to assess the impact of assimilation of the GNSS tomography outputs on the weather forecasts, we compared the base run (BASE; without data assimilation) of the WRF model to two assimilation cases (TUW set1 and WUELS set1). The comparison has been performed for the period of 72-hour heavy precipitation event (2013-05-31 00 UTC - 2013-06-03 00 UTC). The accumulated precipitation has been calculated as a sum of the 6-hour forecasts, starting from the assimilation time. Figure 13 presents the field of precipitation in the WRF domain area.

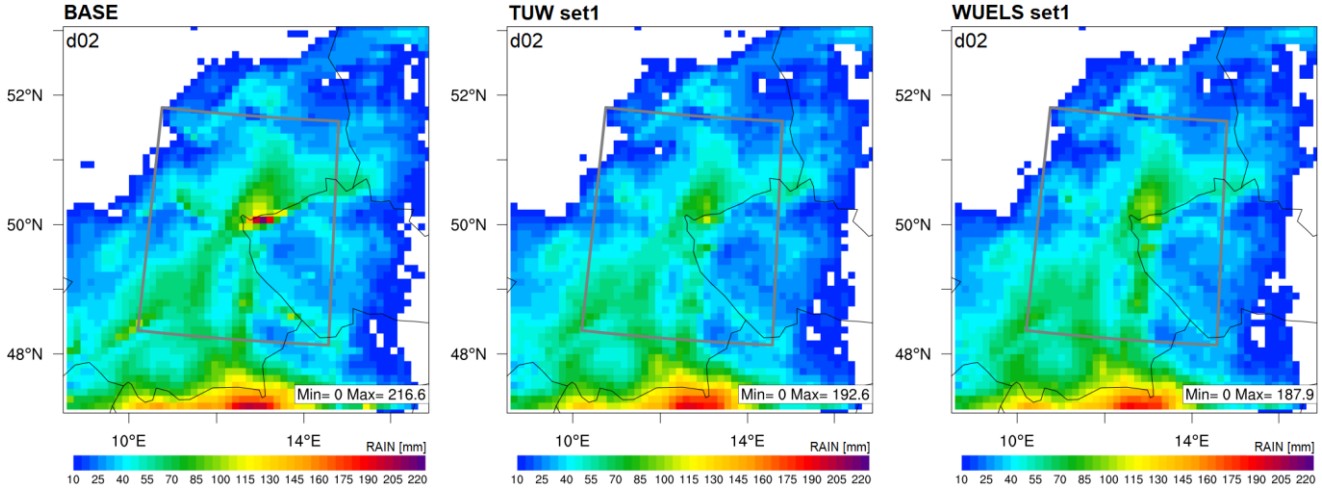

**Figure 13: Total precipitation accumulated for 72 hours (2013-05-31 00 UTC - 2013-06-03 00 UTC) for the WRF model forecasts: BASE (left), TUW set1 (middle), and WUELS set1 (right). The grey line indicates the boundaries of the GNSS tomography models' inner domain.**

In the case of the base run, the highest values of the accumulated precipitation are located in the centre and the south part of the domain area. For both assimilation runs (TUW set1, WUELS set1), the highest precipitation occurs only in the south part of the domain. Comparing the base run to the assimilation runs, we can observe the strong drying impact of the assimilation, especially in the central area, where the values of the precipitation decreased from approx. 220 ppm (base) to approx. 120 ppm (assimilation). The maximal precipitation for the base run is 216.6 mm, whereas this value is noticeably lower for the TUW set1 (192.6 mm) and WUELS set1 (187.9 mm) assimilations. Based on the weather situation (see Section 5), the numbers of the accumulated precipitations are overestimated for all WRF runs. However, the assimilation of the GNSS tomography data decreases the amount of precipitation in the model.

## 7.3 Assimilation results at analysis time

Based on the radiosonde observations (10548, 10771), we calculated the statistics (bias and standard deviation) between the radiosonde data and the model runs at the time of analysis. We analysed three meteorological parameters: Relative Humidity (RH), Temperature (T), and Wind Speed (WS), for the whole tested period (29 May 2013 00:00 UTC – 13 June 2013 18:00 UTC), in the location of radiosonde stations (10548, 10771, 11520). The accuracy of the radiosonde measurements is 5% in terms of the relative humidity, 0.5°C for air temperature, and 1.5 m/s for wind speed (OFCM, 1997). Figure 14 presents the statistics of the differences between radiosonde measurements and model runs as a function of pressure. Relative humidity slightly improved in terms of bias in the boundary layer (900 – 700 hPa) by 1% – 2%. In higher layers, bias of relative humidity increased (by 1% – 5%) but standard deviation decreased (by 1% – 2%) in the case of assimilation of the TUW data (especially within the pressure range of 700 to 400 hPa). Bias of temperature is growing after assimilation in the pressure range of 900 – 400 hPa, whereas in the higher troposphere, the bias is decreased when compared

with the base run, by 0.5°C for WUELS and 0.2°C for TUW. For the wind speed, there is a small impact of data assimilation on the values of bias, but in terms of standard deviation there is a noticeable improvement for pressure range of 600 – 400 hPa (0.2 m s$^{-1}$).

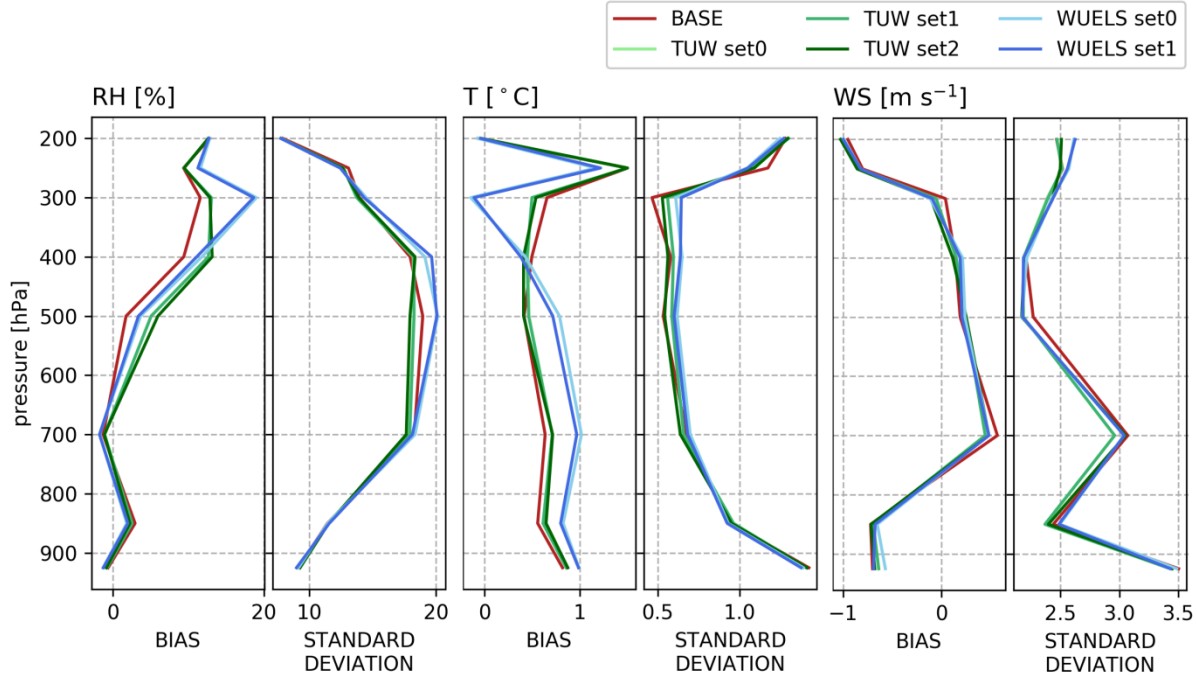

**Figure 14: From the right, statistics (bias, standard deviation) for Relative Humidity (RH), Temperature (T), and Wind Speed (WS) base forecast and model fed with TUW (set0, set1, set2) and WUELS (set0, set1) data when compared with radiosonde measurement; comparison at assimilation time (6 hours since model start).**

### 7.4. Assimilation results at simulation time (short-term forecast)

The impact of assimilation on the short-term forecast has been validated against radiosonde observations. Figure 15 summarizes the statistics (bias, standard deviation) for the individual weather forecasts with and without assimilated tomographic output data – separately for relative humidity, temperature, and wind speed. This comparison has been conducted 6 hours after assimilation (i.e., 12 hours since model start) for the whole tested period, in the location of radiosonde stations (10548, 10771).

In order to assess the impact of the assimilation of the GNSS tomography outputs on the weather forecast, we subtracted the absolute values of statistics for the case of assimilation and the base model run. If the difference between the statistics is negative, it means that the assimilation of the GNSS tomography output decreases the value of bias and/or standard deviations between the model and radiosonde observation. It implies that the model after assimilation becomes closer to the real observations.

In the beginning of the tested period, i.e., during the first two days, the weather in the Central Europe area was rather calm, thus for all assimilation cases the differences between the statistics for the weather forecast after tomography data assimilation and the base run are on similar level. In case of a relative humidity assimilation, in time of the heavy precipitation events (01-03 June 2013), the absolute differences of bias between the models after assimilation of the tomography data and the base run have negative values. The highest positive impact (i.e., decrease of the value of bias) occurs in case of assimilation of the WUELS tomography outputs and exceeds -4.5% for both bias and the standard deviation differences. In case of the TUW GNSS tomography data assimilation, the differences in the statistical numbers reach around -1 % for set2 and around -2.5% for set0 and set1.

We observe that the assimilation of the tomography outputs does not have any significant influence on the other examined meteorological parameters. In case of the temperature, the differences hold the level up to 0.1 °C for both bias and standard deviations. Since 04 June 2013, the tomography wet refractivity assimilation gives mainly a positive effect for the differences (i,e,, the forecast of temperature works better without tomography data assimilation) in case of WUELS data (set0, set1), whereas the differences for the TUW data (set0, set1, set2) are close to zero. The statistics for the wind speed look very similar; the assimilation of the tomography data modifies barely the difference between results of statistics, up to 0.5 m/s for both bias and standard deviation differences.

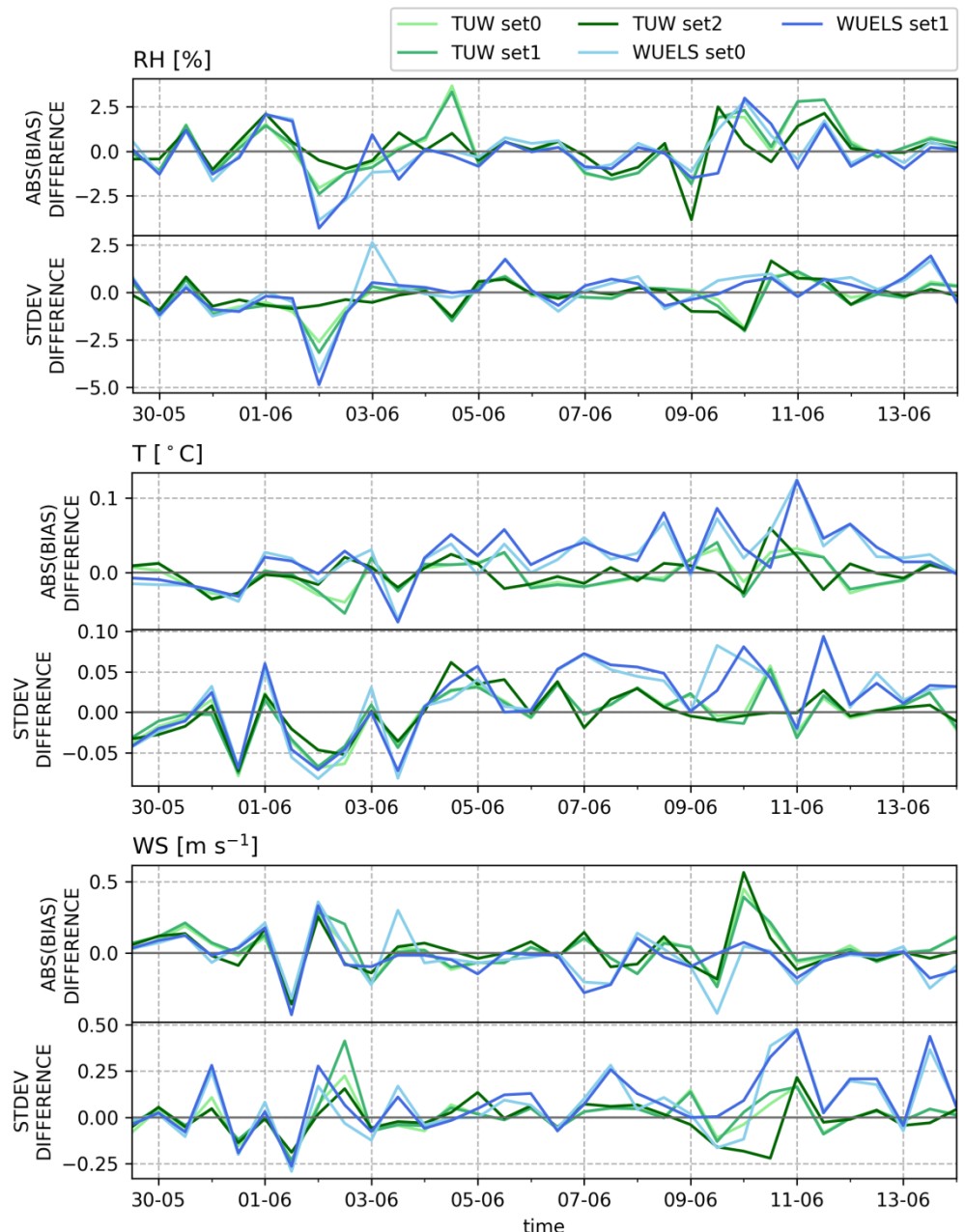

**Figure 15: Absolute bias and standard deviation of the differences between statistics of model fed with tomographic data (TUW (set0, set1, set2) and WUELS (set0, set1)) and base forecast for Relative Humidity (RH), Temperature (T), and Wind Speed (WS) 6 hours after assimilation (12 hours since model start).**

**8 Conclusions**

The GNSS tomography wet refractivity fields can play a key role in the evolution of the weather forecast quality. Although nowadays it is possible to perform GNSS tomography only on regional scale, where the density of the stations is large enough to enable tomography, the outputs provide the crucial information about a local water vapour horizontal and

vertical distribution. In this study, GNSS tomography was performed by two models (TUW, WUELS), which are based on different tomography principles. We analysed the data for the area of Central Europe in the period of 29 May – 14 June 2013, when heavy precipitation events were observed. The SWDs were calculated based on estimates of the ZTDs and horizontal gradients, provided for 88 GNSS sites by Geodetic Observatory Pecny (GOP). For the TUW model, three sets of SWD observations were tested (set0 without compensation for hydrostatic anisotropic effects, set1 with compensation of this

effect, set2 cleaned by wet delays outside the inner voxel model), whereas for the WUELS model the set0 and set1 were analysed.

The validation of the tomography results with radiosonde data shows that due to proper weighting the tomography solution can correct deficits in the a priori model. Two different approaches of elimination of the outer parts of SWD observations, which do not pass through the model domain, were examined. The use of the outer model domain led to

similar results as a removal of the outer SWD parts using ray-tracing technique.

In order to assess the benefits of GNSS tomography outputs on weather forecasting, we arranged five assimilation tests. This process is enabled by the use of the WRF GPSREF operator for which the hydrostatic part of refractivity was computed using ALADIN-CZ forecast data and added to the GNSS tomography field of wet refractivity. During the assimilation process a lot of observations, depending on the observation level, were rejected. The differences between the

GNSS tomography observations from both models (TUW, WUELS) and the background data in the lower part of the troposphere were significant. Therefore, most of the observations were assigned as incorrect during the quality check procedure and they do not have any impact on the assimilation results. In the higher part of the troposphere, in case of the TUW model, most of the observations have been successfully assimilated, whereas in case of the WUELS data about 1000 observations have been omitted at every height layer above 6 km. The accuracy of the tomography model outputs was

determined by the comparison with RS observations. The validation indicated large variations in the WUELS solution, especially in the upper part of the troposphere. Hence, the verification process in the assimilation is consistent with the quality of the data and we conclude that the quality check system dedicated to radio occultation data can be applied for the assimilation of tomography outputs.

Within the study period, assimilation was performed under diverse weather conditions. The heaviest precipitation

occurred in the period of 01-03 June 2013. During this period, the most significant positive effect after assimilation of tomography data was noticed. The 72-hour accumulated total precipitation during the heavy precipitation event was overestimated in the base run of the model, however, after assimilation of the GNSS tomography data a drying effect could be observed. Comparing to the radiosonde observations, the weather forecast in the period of severe weather was improved

in terms of relative humidity (bias and standard deviation) and temperature (standard deviations), whereas no impact was observed in terms of wind speed. However, statistically more robust results are expected from a long-term assimilation campaign. Hereby, the advantage of using the GNSS tomography data in the weather forecasting could be verified against the assimilation of other GNSS tropospheric parameters, e.g., ZTD or IWV. Besides, for assimilation using WRF GPSREF operator the hydrostatic part of refractivity might be calculated from the background model (at time of assimilation) instead of ALADIN-CZ, in order to avoid influences caused by differences between the two models (ALADIN-CZ and WRF) on the results. Future research will cover the development of the observation operator dedicated to the assimilation of the GNSS tomography wet refractivity, in order to eliminate the need of an external data source to derive the hydrostatic part of the refractivity.

## Data availability

The data set was primarily collected for the purpose of the COST Action ES1206 (GNSS4SWEC project). The data are available as given in Dousa et al. (2016).

## Author contribution

Natalia Hanna as a first and corresponding author administered the project, formulated the methodology of assimilation process, investigated the assimilation experiments, and validated the results. Estera Trzcina, Gregor Moeller, and Witold Rohm prepared the methodology of GNSS tomography process. Estera Trzcina contributed by investigation of GNSS tomography for WUELS model, and conducted the formal analyses in the GNSS tomography and data assimilation part, and validated the results. Gregor Moeller contributed by investigation of GNSS tomography for TUW model, conducted the formal analyses in the GNSS tomography part, and validated the results. Witold Rohm and Robert Weber initiated the research idea, supervised the project, and reviewed the manuscript. Natalia Hanna and Estera Trzcina prepared the manuscript with contributions from all co-authors.

## Acknowledgements

This study has been organized within the EU COST action ES1206 (GNSS4SWEC). The research has been supported by the Polish National Science Centre (project no. UMO-2015/17/B/ST10/03827) and the Wroclaw Center of Networking and Supercomputing (http://www.wcss.wroc.pl/) computational grant using MATLAB Software License No: 101979. The authors thank all the institutions that provided data for the campaign: GNSS data from the Geodetic Observatory Pecny (GOP; http://www.pecny.cz/), NWM data from the local area model ALADIN-CZ provided by the Czech

Hydrometeorological Institute (CHMI; http://portal.chmi.cz), and radiosonde observations from the Deutscher Wetterdienst (DWD Germany, https://www.dwd.de).

**Competing interests**

The authors declare that they have no conflict of interest.

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
