# Peer review of "Assimilation of GNSS tomography products into WRF using radio occultation data assimilation operator"

_Atmospheric Measurement Techniques, 2018_

## Referee Comment (RC1) · Anonymous Referee #1 · 15 Mar 2019

The authors present the assimilation of GNSS tomography products into WRF during the period when heavy precipitation events occurred. Two tomography models and three SWD sets were tested. Results of the GNSS tomography data assimilation were validated by radiosonde measurements in terms of relative humidity, temperature and wind.

In general content of the article is abundant and well written, however, there are some points that the authors should consider before it would be suitable for publication.

Remarks:

1) The time resolution of the tomographic results has not been clearly indicated in the

paper. In line 5 of page 4, the ZTD estimates have a 1 h time resolution. In line 7 of page 13, it shows the solutions have a 6 h resolution. It is not clear how long of the SWD data are stacked for each tomographic solution. Under extreme weather conditions, the water vapor changes quickly thus a reasonable resolution is very important. 2) Three sets of SWD observations were tested: set0 without compensation for hydrostatic anisotropic effects, set1 with compensation of this effect and set2 cleaned by wet delays outside the inner voxel model. First, why not test the set2 by also considering the compensation of hydrostatic anisotropic effects. Another concern is that why not test set2 for WUELS model? 3) In the voxel discretization, authors divide the region into an inner voxel and an out voxel. The outer voxel is used to also include those signals penetrate the model from the laterals. However, authors should explain how to model the SWDs in the outer voxel. As seen in Figure 3, it seems the outer voxels are too coarse to model the SWDs. 4) Line 4 of page 7, how did you get the number of 120 times in the quality control? 5) For Figures 8, 9, since the wet refractivity varies greatly over the time and space. It is not convenient to compare your results with previous studies. I thus suggest authors to also give the statistics of relative RMS.

---

## Referee Comment (RC2) · Anonymous Referee #3 · 14 May 2019

In their paper, the authors discuss the assimilation of GNSS tomography products from two software (and different output variants) into the WRF NWP model during the severe precipitation event occurring in central Europe (29 May – 14 June 2013). This event is described in the paper Dousa et al. 2016, which presents the COST Action ES1206 (GNSS4SWEC) benchmark campaign. The authors uses the datasets (GNSS tropospheric products, NWP model data...) from that benchmark campaign as input for their study, either as input for their tomography or for comparing/validating their results.

The paper is generally well structured and well written. However, some points - listed here below - requires clarification, additional information, and/or improvements:

[Figure]

- Acronyms for the dataset and model outputs: it is recommended to use - throughout the paper – acronyms that clearly identify the dataset (e.g. set1 → ANHYDRO_COR) or the model output (e.g. TUW solution 1 –> TUW1).

- At several occasions, the paper somehow lack of providing reference numbers (e.g. requirements or typical uncertainties of measurements) that are necessary to properly interpret the findings. Please add them.

- One may ask himself why to carry out tomography on SWD, then assimilate tomography output in NWP models, instead of directly assimilating SWD in NWP models. Assimilating directly SWD in NWP is also one step less in the processing chain and so might be faster for operational purpose. I think a short paragraph in the introduction on this point would be an added value to the paper. Please elaborate on this.

- The authors mentions two assimilation operators. However it is not clear what are the difference between both and why they chosen GPSREF for their study. Please clarify this point.

- The horizontal resolution of the outer voxels are quite coarse for the type of phenomena targeted. Would it have improved the results if it was finer? If yes, why not having applied a finer resolution in the outer voxel? Also why not having included SWD from GNSS station in the outer voxels? (The benchmark campaign includes plenty of GNSS stations data in that zone).

- In the tomography, the authors carry out residuals screening. This is set to 120 times the RMS in the TUW case while it is set to 3 times the standard deviation in the WUELS case. Please indicate how you decided on these numbers and why they look quite relaxed in the TUW case compared to the WUELS case. (See also detailed comments).

- Several figures and tables can be improved (see detailed comments).

- Please mention why you chose to use the ALADIN-CZ (from the benchmark campaign) for some part of the paper instead of using WFR model outputs (which sounds more consistent).

- The section on "intra-technique comparisons" can be completed by discussing all comparisons (see detailed comments) and providing more clear conclusions at the end of the section.

- The section on "diagnosis output" is missing some information and should ends with clear interpretation and conclusions (see also detailed comments).

- The section "7.3 assimilation output results at simulation time": I would recommend the authors to first start with a full comparison between the model results without assimilation and with assimilation over the complete domain (i.e. not restricting to the radiosonde location). This would first give an indication of the impact of assimilating the tomography output. This can take to form of typical skills and scores used in NWP impact studies or some statistical values along with a box plots of certain essentials variables (temperature, relative humidity...). Also in the paper, please mention which exact temperature field you extract from the model. Then, in second step, you can add the radiosonde comparison and some interpretation of potential biases...

- For the data assimilation, please mention if you have carry out any bias correction (typical in NWP data assimilation). If not: why not? If yes: please describe.

Detailed Comments can be found in the supplementary material.

Please also note the supplement to this comment:
https://www.atmos-meas-tech-discuss.net/amt-2018-419/amt-2018-419-RC2-supplement.zip

---

## Author Comment (AC2) · 11 Jun 2019

First of all, we appreciate a lot such valuable and detailed Review! Attached are 3 documents:

1) Review2_answers.pdf - authors answers to the questions; 2) amt-2018-419-supplement_with_authors_answers.docx - supplement with the detailed comments provided by Reviewer 2, together with authors' answers; 3) Hanna_et_al_2018_update.pdf - reviewed (updated) version of the manuscript.

We hope that we have clarified all the issues.

[Figure]

Please also note the supplement to this comment:
https://www.atmos-meas-tech-discuss.net/amt-2018-419/amt-2018-419-AC2-supplement.zip

―――――――――――――――――――

---

## Author Response (AR1)

1) **The time resolution of the tomographic results has not been clearly indicated in the paper. In line 5 of page 4, the ZTD estimates have a 1 h time resolution. In line 7 of page 13, it shows the solutions have a 6 h resolution. It is not clear how long of the SWD data are stacked for each tomographic solution. Under extreme weather conditions, the water vapor changes quickly thus a reasonable resolution is very important.**

The time resolution of the tomographic results is indicated in table 3 (Time settings: every 6 hours). The ZTD estimates have 1 h time resolution, however, because of our assimilation settings, we needed the tomography outputs only from 00, 06, 12, and 18 UTC. The GNSS tomography models have been run every 6 hours and the SWD were not stacked. It has been written more explicitly in Section 2 of the new version of the paper.

2) **Three sets of SWD observations were tested: set0 without compensation for hydrostatic anisotropic effects, set1 with compensation of this effect and set2 cleaned by wet delays outside the inner voxel model. First, why not test the set2 by also considering the compensation of hydrostatic anisotropic effects. Another concern is that why not test set2 for WUELS model?**

In the case of set2 anisotropic effects were not compensated. It is right that in the comparison between set1 and set2 not only the effect of the outer delay but also of the hydrostatic asymmetry is shown. However, to fix this we have redrawn Figure 5 (4 in the new numeration) and 9 by replacing set1 with set0, to show only the effect of the outer delay.

In the case of the WUELS model, the coordinates of the voxels are projected to the UTM coordinate system. In set2, the ray-tracing through the ALADIN-CZ data was made using the ellipsoidal (BLH) coordinates. As the TUW model uses the same coordinate system (BLH), the application of the ray-traced data was possible without any modifications in the tomography model. Because of the complications in using the ray-traced data in the WUELS model (deformations caused by the coordinates transformation), we decided to test this approach only for the TUW model. In our future work, we plan to adjust the WUELS model to operate on the ellipsoidal coordinate system.

3) **In the voxel discretization, authors divide the region into an inner voxel and an out voxel. The outer voxel is used to also include those signals penetrate the model from the laterals. However, authors should explain how to model the SWDs in the outer voxel. As seen in Figure 3, it seems the outer voxels are too coarse to model the SWDs.**

In the case of set0 and set1, the refractivity in the outer domain is estimated together with the refractivity in the inner voxel domain. In the case of set2, the paths length and refractivity in the outer domain is set to zero (since already removed beforehand). The outer domain is coarse to avoid passing signals through lateral boundaries. This approach has some drawbacks as the signal is considered to be a straight line over the long distance in the outer voxels. This led us to apply two approaches for removing the outer parts of SWD (set0 and set1, set2) and compare them.

4) **Line 4 of page 7, how did you get the number of 120 times in the quality control?**

The number of 120 was defined empirically and removes large outliers in SWD only.

5) **For Figures 8, 9, since the wet refractivity varies greatly over the time and space. It is not convenient to compare your results with previous studies. I thus suggest authors to also give the statistics of relative RMS**

Thank you for this comment; we have redrawn Figures 8, 9 to include also the statistics of RMS.

1. **Acronyms for the dataset and model outputs: it is recommended to use – throughout the paper – acronyms that clearly identify the dataset (e.g. set1 - ANHYDRO_COR) or the model output (e.g. TUW solution 1 - TUW1).**

   As the acronyms that clearly identify the model output would be very long (e.g. WUELS_NO_ANHYDRO_OUTER), we decided to leave the acronyms of the GNSS tomography models (TUW, WUELS) and the number of the SWD solution (set0, set1, set2). In Table 3, the detailed characteristics of the particular sets is presented. In the previous version of the paper, some inconsistency occurred (e.g. "sol" instead of "set"), so we have corrected it.

2. **At several occasions, the paper somehow lack of providing reference numbers (e.g. requirements or typical uncertainties of measurements) that are necessary to properly interpret the findings. Please add them.**

   The paper has been modified according to the suggestions given in the supplement to the Review.

3. **One may ask himself why to carry out tomography on SWD, then assimilate tomography output in NWP models, instead of directly assimilating SWD in NWP models. Assimilating directly SWD in NWP is also one step less in the processing chain and so might be faster for operational purpose. I think a short paragraph in the introduction on this point would be an added value to the paper. Please elaborate on this.**

   A short paragraph on the STD data assimilation has been added to the introduction part of the paper.

   The challenging part related to developing the operator for SWD/STD is high computation costs of each of the parts: Forward Operator, Tangent Linear and Adjoint.
   Forward Operator requires computing for each SWD/STD observation raytraced delay between satellite and ground-based receiver. Another issue is the high nonlinearity of the signal path for low satellites and in result parametrisation for Tangent Linear operator is complex problem and requires number of iterations. On top of that the Adjoint operator requires interpolating the increments to number nodes around signal path which is computationally expensive as the signal traverse through the number of voxels in arbitrary direction (not typical vertical). Moreover, similar as with Integrated Water Vapour and Zenith Troposphere Delay assimilation, the observation uncertainty is difficult to assign, as the STD/SWD are point observations integrated over certain part of the model with different quality of retrieval in different parts of the model.
   In contrast GNSS tomography outputs are profile observations, similar to the operators already developed and operationally used in weather models i.e. Radio Occultation (Healy, 2007). It was demonstrated in number of studies that these profiles increase the quality of model fields (Cucurull et al., 2007; Poli et al., 2010; Buontempo et al., 2009; Healy, 2008). The quality of refractivity profiles are relatively easy to obtain by, e.g., comparison to the radiosonde profiles, and assigning proper uncertainty to the coinciding levels (Brenot et al., 2018). It is our long range aim to develop similar to GPSREF operator for tomography outputs with uncertainties and quality control tailor-made for tomography observations. However, this is out of the scope of present study.

   Buontempo, C., Jupp, A., and Rennie, M. (2008). Operational NWP assimilation of GPS radio occultation data. Atmospheric Science Letters, 9(3), 129-133.
   Brenot, H., Rohm, W., Kačmařík, M., Möller, G., Sá, A., Tondaś, D., Rapant, L., Biondi, R., Manning, T., and Champollion, C. (2018). Cross-validation of GPS tomography models and methodological improvements using CORS network, Atmos. Meas. Tech. Discuss., https://doi.org/10.5194/amt-2018-292, in review.
   Cucurull, L., Derber, J. C., Treadon, R., and Purser, R. J. (2007). Assimilation of global positioning system radio occultation observations into NCEP's global data assimilation system. Monthly weather review, 135(9), 3174-3193.

Healy, S. B. (2007). Operational assimilation of GPS radio occultation measurements at ECMWF. ECMWF Newsletter, 111, 6-11.

Healy, S. B. (2008, June). Assimilation of GPS radio occultation measurements at ECMWF. In Proceedings of the GRAS SAF Workshop on Applications of GPSRO measurements, ECMWF, Reading, UK (pp. 16-18).

Poli, P., Healy, S. B., & Dee, D. P. (2010). Assimilation of Global Positioning System radio occultation data in the ECMWF ERA–Interim reanalysis. Quarterly Journal of the Royal Meteorological Society, 136(653), 1972-1990.

4. **The authors mentions two assimilation operators. However it is not clear what are the difference between both and why they chosen GPSREF for their study. Please clarify this point.**

In the introduction part, the study by Trzcina and Rohm (2019) has been mentioned. In that study, the GNSS tomography output was assimilated into the WRF model using an operator dedicated to radio occultation (RO) observations of total refractivity (GPSREF). It is the same operator that has been used in our study. As the name of the operator has not been stated in this part of our introduction section, it could be misleading. To make it clear, the name of the operator used by Trzcina and Rohm (2019) has been added.

5. **The horizontal resolution of the outer voxels are quite coarse for the type of phenomena targeted. Would it have improved the results if it was finer? If yes, why not having applied a finer resolution in the outer voxel? Also why not having included SWD from GNSS station in the outer voxels? (The benchmark campaign includes plenty of GNSS stations data in that zone).**

The outer domain is coarse to avoid passing signals through lateral boundaries. This approach has some drawbacks as the signal is considered to be a straight line over the long distance in the outer voxels. This led us to apply two approaches for removing the outer parts of SWD (set0 and set1, set2) and compare them.

The stations in the outer domain have not been used, as the main purpose of this approach is to remove the outer parts of the GNSS signal from the tomographic solution. The observations from the stations located in the outer voxels are rarely passing through the inner domain, thus we expect that using them would not help to improve the quality of the tomographic solution.

6. **In the tomography, the authors carry out residuals screening. This is set to 120 times the RMS in the TUW case while it is set to 3 times the standard deviation in the WUELS case. Please indicate how you decided on these numbers and why they look quite relaxed in the TUW case compared to the WUELS case. (See also detailed comments).**

In the TUW solution the RMS of the weighted residuals ($r_{sum}$) is computed as follows:

$$r_{sum} = \sqrt{\frac{(res' \cdot P \cdot res)}{length(res)}}$$

where 'res' are the post-fit residuals and 'P' is the weighting matrix for both, a priori data and observations. After each iteration, post-fit residuals (only applied to SWDs) larger than $120*r_{sum}$ are treated as outliers and removed from the processing. The threshold of 120 was found empirically and allows for removing large outliers (usually $< 2\%$ of SWDs at low elevation angle).

In the WUELS solution, the standard deviation of the residuals ($r_{sum}$, calculated the same way as in the TUW case) was multiplied by 3. This was set as the threshold for the outlying observations. It was found empirically and allows for removing about 4% of SWDs, mainly at low elevation angles.

This information has been included in the revised version of the manuscript (Sections 3.1 and 3.2).

7. **Several figures and tables can be improved (see detailed comments).**

The figures and tables have been improved according to the detailed comments given in the supplement to the Review.

8. **Please mention why you chose to use the ALADIN-CZ (from the benchmark campaign) for some part of the paper instead of using WFR model outputs (which sounds more consistent).**

Main reason behind using ALADIN-CZ is that the tomography studies show that the impact of well-defined a priori data (Brenot et al., 2018), so the tomography model could be mostly used to resolved inconsistencies in the model, mostly these are related to vertical inversion, not detected by models. Moreover, ALADIN-CZ was used through the COST Action GNSS4SWEC troposphere studies related to GNSS applications in meteorology (Kacmarik et al., 2017; Dousa et al., 2016) to stay consistent with the programme wide adoption of the NWP model we decided to use ALADIN-CZ. We are aware that using ALADIN-CZ as an apriori in tomography has an impact on the final tomography retrieval and is transpiring into the assimilated refractivities, however to fully study this impact we would need to construct tomography operator. Using this tool, we could assimilate ALADIN-CZ based wet refractivities into the WRF model. However, this is out of the scope of this paper and is a subject of another study.

Brenot, H., Rohm, W., Kačmařík, M., Möller, G., Sá, A., Tondaś, D., Rapant, L., Biondi, R., Manning, T., and Champollion, C. (2018). Cross-validation of GPS tomography models and methodological improvements using CORS network, Atmos. Meas. Tech. Discuss., https://doi.org/10.5194/amt-2018-292, in review.
Douša, J., Dick, G., Kačmařík, M., Brožková, R., Zus, F., Brenot, H., Stoycheva, A., Möller, G., and Kaplon, J. (2016). Benchmark campaign and case study episode in central Europe for development and assessment of advanced GNSS tropospheric models and products, Atmospheric Measurement Techniques, 9, 2989-3008, https://doi.org/10.5194/amt-9-2989-2016.
Kacmarík, M., Douša, J., Dick, G., Zus, F., Brenot, H., Möller, G., Pottiaux E., Kaplon J., Hordyniec P., Václavovic P. and Morel, L. (2017). Inter-technique validation of tropospheric slant total delays, Atmospheric Measurement Techniques, 10(6), doi:10.5194/amt-10-2183-2017.

9. **The section on "intra-technique comparisons" can be completed by discussing all comparisons (see detailed comments) and providing more clear conclusions at the end of the section.**

Paragraph 6.1 has been updated according to the suggestions of the Reviewer. The comparison between set0 and set1 has been presented for the TUW model (Figure 3) and discussed for the WUELS model as well. In the case of set0, the comparison of statistics (bias, standard deviation) for both GNSS tomography models (TUW, WUELS) are looking very similar to the comparison TUW set1 versus WUELS set1 (because of small discrepancies between set0 and set1 in both models, which is presented in the new Fig. 3 and discussed in the updated version of the text).

According to Revision 1, it is recommended to omit the comparison between TUW set1 and TUW set2 (it shows not only differences caused by different approaches in removing the outer parts of SWD, but also impact of anisotropy correction). Instead of this comparison, we compared TUW set0 versus TUW set2 (here only the effect of two approaches in removing outer delays is presented).
The conclusions at the end of the section 6.1 have been provided.

10. **The section on "diagnosis output" is missing some information and should ends with clear interpretation and conclusions (see also detailed comments).**

The section has been updated according to the detailed comments.

11. **The section "7.3 assimilation output results at simulation time": I would recommend the authors to first start with a full comparison between the model results without assimilation and with**

**assimilation over the complete domain (i.e. not restricting to the radiosonde location). This would first give an indication of the impact of assimilating the tomography output. This can take to form of typical skills and scores used in NWP impact studies or some statistical values along with a box plots of certain essentials variables (temperature, relative humidity...). Also in the paper, please mention which exact temperature field you extract from the model. Then, in second step, you can add the radiosonde comparison and some interpretation of potential biases...**

Since the assimilation of the GNSS tomography outputs has the highest impact on the relative humidity and much smaller on the temperature and the wind speed, we assumed that the 72-hour accumulated total precipitation is the best indicator to show the impact of assimilation over the complete model domain. We have presented the precipitation field for the base run and the results of assimilation runs (TUW set1, WUELS set1) during the heavy precipitation event (2013-05-31 00 UTC – 2013-06-03 00 UTC) in Fig. 12, together with the tomography domain. This comparison shows that the assimilation changes the precipitation field mainly within the area of the GNSS tomography. Besides, the assimilation of the GNSS tomography data has the drying effect on the model.

According to the suggestions, we added a new comparison between the base run and assimilation runs (TUW set1, WUELS set1) in terms of the accumulated total precipitation over the full domain (paragraph 7.2 in the updated manuscript; Fig. 12).
The air temperature field was derived from the model using the NCL (NCAR Command Language) function (wrf_user_getvar) in degree Celsius.

12. **For the data assimilation, please mention if you have carry out any bias correction (typical in NWP data assimilation). If not: why not? If yes: please describe.**

The bias correction was not applied in this study, there are two reasons behind. Firstly, in our previous study (Trzcina and Rohm, 2019) we initially were calculating the mean differences between tomography observations and radiosonde on three sites across Poland twice a day and then try to extrapolate these differences across model and between observation epochs, however the results were negative. The impact of assimilation of GNSS tomography bias-corrected data were lower than bias-uncorrected. Secondly, the bias of tomography model is not static and varies between epochs dramatically and is most likely complex function of a priori data quality, number of observations, satellites and elevation angles. The tomography error variation over time and space is a subject of another study that is currently run by IAG Tomography Working Group
https://iag.dgfi.tum.de/fileadmin/IAG-docs/Travaux2017/04_Commission_4_2015-2017.pdf

[revised manuscript text omitted]

---

## Author Response (AR2)

Minor revisions and comments

1. **Page 5, lines 7-8: The authors answered my question why they projected in the direction of both GPS and GLONASS satellites while Dousa et al. only processed GPS observations. However, the manuscript does not mention it. Please mention at least the fact that Dousa et al. have processed only GPS observations but that you can project in the direction of all satellites from both constellation because you expect that the error will not be significant.**

   The information about projection in the direction of both GPS and GLONASS satellites has been included in the revised version of the manuscript (Page 5, lines 10-12).

2. **Page 10, line 9: "This information exchange between the grids in both directions" --> this sentence is not correct, do you mean something like "The latest allow information exchange between the grids in both directions"?**

   This sentence has been revised according to the suggestion of the Reviewer (Page 10, lines 18).

3. **Page 15, line 15: "larger standard deviations" --> too vague, please be more precise, e.g. "larger standard deviations up to …".**

   The information about the value of standard deviations has been included in the revised version of the manuscript, following the suggestion of the Reviewer (Page 15, line 15).

4. **Page 17, line 17: "horizontally interpolated" --> by which method, please be concrete.**

   Wet refractivity values have been linearly interpolated, this information has been added in the revised version of the manuscript (Page 17, line 17).

5. **Page 18, Table 4 (recommendation): it would have been good to provide information about the extremes values. This could be easily added with a box plot graph (for ALL) for each 3 comparisons.**

   Following the recommendation of the Reviewer, the box plot graph for each 3 comparisons has been provided in order to present the information about outliers (Fig. 7). Additionally, a short description to this Figure is given in Page 19, lines 8-11.

Technical comments

All the technical comments have been applied, following the suggestions of the Reviewer.

[revised manuscript text omitted]